# AI-SARAH: Adaptive and Implicit Stochastic Recursive Gradient Methods

## Abstract

We present *AI-SARAH*, a practical variant of *SARAH*. As a variant of *SARAH*, this algorithm employs the stochastic recursive gradient yet adjusts step-size based on local geometry. *AI-SARAH* implicitly computes step-size and efficiently estimates local Lipschitz smoothness of stochastic functions. It is fully adaptive, tune-free, straightforward to implement, and computationally efficient. We provide technical insight and intuitive illustrations on its design and convergence. We conduct extensive empirical analysis and demonstrate its strong performance compared with its classical counterparts and other state-of-the-art first-order methods in solving convex machine learning problems.

## 1 Introduction

We consider the unconstrained finite-sum optimization problem

$$\min_{w \in \mathcal{R}^d} [P(w) \overset{\text{def}}{=} \tfrac{1}{n} \textstyle\sum_{i=1}^n f_i(w)]. \tag{1}$$

This problem is prevalent in machine learning tasks where $w$ corresponds to the model parameters, $f_i(w)$ represents the loss on the training point $i$, and the goal is to minimize the average loss $P(w)$ across the training points. In machine learning applications, (1) is often considered the loss function of Empirical Risk Minimization (ERM) problems. For instance, given a classification or regression problem, $f_i$ can be defined as logistic regression or least square by $(x_i, y_i)$ where $x_i$ is a feature representation and $y_i$ is a label. Throughout the paper, we assume that each function $f_i$, $i \in [n] \overset{\text{def}}{=} \{1, ..., n\}$, is smooth and convex, and there exists an optimal solution $w^*$ of (1).

### 1.1 Main Contributions

We propose *AI-SARAH*, a practical variant of stochastic recursive gradient methods (Nguyen et al., 2017) to solve (1). This practical algorithm explores and adapts to local geometry. It is adaptive at full scale yet requires zero effort of tuning hyper-parameters. The extensive numerical experiments demonstrate that our tune-free and fully adaptive algorithm is capable of delivering a consistently competitive performance on various datasets, when comparing with *SARAH*, *SARAH+* and other state-of-the-art first-order method, all equipped with fine-tuned hyper-parameters (which are selected from $\approx 5,000$ runs for each problem). This work provides a foundation on studying adaptivity (of stochastic recursive gradient methods) and demonstrates that a **truly adaptive stochastic recursive algorithm can be developed in practice.**

### 1.2 Related Work

Stochastic gradient descent (*SGD*) (Robbins & Monro, 1951; Nemirovski & Yudin, 1983; Shalev-Shwartz et al., 2007; Nemirovski et al., 2009; Gower et al., 2019) is the workhorse for training supervised machine learning problems that have the generic form (1).

In its generic form, *SGD* defines the new iterate by subtracting a multiple of a stochastic gradient $g(w_t)$ from the current iterate $w_t$. That is,

$$w_{t+1} = w_t - \alpha_t g(w_t).$$

In most algorithms, $g(w)$ is an unbiased estimator of the gradient (i.e., a stochastic gradient), $\mathbb{E}[g(w)] = \nabla P(w), \forall w \in \mathcal{R}^d$. However, in several algorithms (including the ones from this paper),

$g(w)$ could be a biased estimator, and convergence guarantees can still be well obtained.

**Adaptive step-size selection.** The main parameter to guarantee the convergence of *SGD* is the *step-size*. In recent years, several ways of selecting the step-size have been proposed. For example, an analysis of *SGD* with constant step-size ($\alpha_t = \alpha$) or decreasing step-size has been proposed in Moulines & Bach (2011); Ghadimi & Lan (2013); Needell et al. (2016); Nguyen et al. (2018); Bottou et al. (2018); Gower et al. (2019; 2020b) under different assumptions on the properties of problem (1).

More recently, *adaptive / parameter-free* methods (Duchi et al., 2011; Kingma & Ba, 2015; Bengio, 2015; Li & Orabona, 2018; Vaswani et al., 2019; Liu et al., 2019a; Ward et al., 2019; Loizou et al., 2020) that adapt the step-size as the algorithms progress have become popular and are particularly beneficial when training deep neural networks. Normally, in these algorithms, the step-size does not depend on parameters that might be unknown in practical scenarios, like the smoothness parameter or the strongly convex parameter.

**Random vector $g(w_t)$ and variance reduced methods.** One of the most remarkable algorithmic breakthroughs in recent years was the development of variance-reduced stochastic gradient algorithms for solving finite-sum optimization problems. These algorithms, by reducing the variance of the stochastic gradients, are able to guarantee convergence to the exact solution of the optimization problem with faster convergence than classical *SGD*. In the past decade, many efficient variance-reduced methods have been proposed. Some popular examples of variance reduced algorithms are *SAG* (Schmidt et al., 2017), *SAGA* (Defazio et al., 2014), *SVRG* (Johnson & Zhang, 2013) and *SARAH* (Nguyen et al., 2017). For more examples of variance reduced methods, see Defazio (2016); Konečný et al. (2016); Gower et al. (2020a); Khaled et al. (2020); Horváth et al. (2020).

Among the variance reduced methods, *SARAH* is of our interest in this work. Like the popular *SVRG*, *SARAH* algorithm is composed of two nested loops. In each outer loop $k \geq 1$, the gradient estimate $v_0 = \nabla P(w_{k-1})$ is set to be the full gradient. Subsequently, in the inner loop, at $t \geq 1$, a biased estimator $v_t$ is used and defined recursively as

$$v_t = \nabla f_i(w_t) - \nabla f_i(w_{t-1}) + v_{t-1}, \tag{2}$$

where $i \in [n]$ is a random sample selected at $t$.

A common characteristic of the popular variance reduced methods is that the step-size $\alpha$ in their update rule $w_{t+1} = w_t - \alpha v_t$ is constant (or diminishing with predetermined rules) and that depends on the characteristics of problem (1). An exception to this rule are the variance reduced methods with Barzilai-Borwein step size, named *BB-SVRG* and *BB-SARAH* proposed in Tan et al. (2016) and Li & Giannakis (2019) respectively. These methods allow to use Barzilai-Borwein (*BB*) step size rule to update the step-size once in every epoch; for more examples, see Li et al. (2020); Yang et al. (2021). There are also methods proposing approach of using local Lipschitz smoothness to derive an adaptive step-size (Liu et al., 2019b) with additional tunable parameters or leveraging *BB* step-size with averaging schemes to automatically determine the inner loop size (Li et al., 2020). However, these methods do not fully take advantage of the local geometry, and **a truly adaptive algorithm: adjusting step-size at every (inner) iteration and eliminating need of tuning any hyper-parameters, is yet to be developed in the stochastic variance reduced framework.** This is exactly the main contribution of this work, as we mentioned in previous section.

## 2 MOTIVATION

With our primary focus on the design of a stochastic recursive algorithm with adaptive step-size, we discuss our motivation in this chapter.

A standard approach of tuning the step-size involves the painstaking grid search on a wide range of candidates. While more sophisticated methods can design a tuning plan, they often struggle for efficiency and/or require a considerable amount of computing resources.

More importantly, tuning step-size requires knowledge that is not readily available at a starting point $w_0 \in \mathcal{R}^d$, and choices of step-size could be heavily influenced by the curvature provided $\nabla^2 P(w_0)$. *What if a step-size has to be small due to a "sharp" curvature initially, which becomes "flat" afterwards?*

To see this is indeed the case for many machine learning problems, let us consider logistic regression for a binary classification problem, i.e., $f_i(w) = \log(1 + \exp(-y_i x_i^T w)) + \frac{\lambda}{2}\|w\|^2$, where $x_i \in \mathcal{R}^d$ is a feature vector, $y_i \in \{-1, +1\}$ is a ground truth, and the ERM problem is in the form of (1). It is

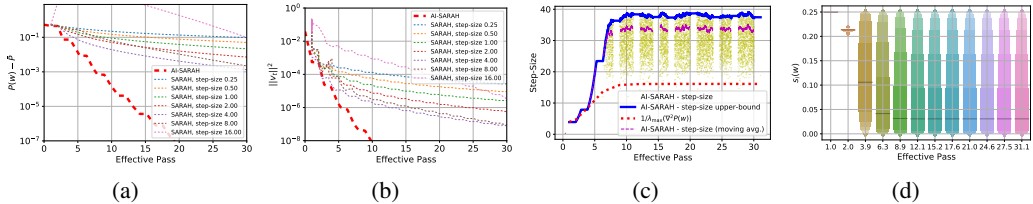

Figure 1: *AI-SARAH* vs. *SARAH*: **(a)** evolution of the optimality gap $P(w) - \bar{P}$ and **(b)** the squared norm of stochastic recursive gradient $\|v_t\|^2$; *AI-SARAH*: **(c)** evolution of the step-size, upper-bound, local Lipschitz smoothness and **(d)** distribution of $s_i$ of stochastic functions. *Note: in (a), $\bar{P}$ is a lower bound of $P(w^*)$; in (c), the white spaces suggest full gradient computations at outer iterations; in (d), bars represent medians of $s_i$'s.*

easy to derive the local curvature of $P(w)$, defined by its Hessian in the form

$$\nabla^2 P(w) = \frac{1}{n}\sum_{i=1}^{n} \underbrace{\frac{\exp(-y_i x_i^T w)}{[1+\exp(-y_i x_i^T w)]^2}}_{s_i(w)} x_i x_i^T + \lambda I. \tag{3}$$

Given that $\frac{a}{(1+a)^2} \leq 0.25$ for any $a \geq 0$, one can immediately obtain the global bound on Hessian, i.e. $\forall w \in \mathcal{R}^d$ we have $\nabla^2 P(w) \preceq \frac{1}{4}\frac{1}{n}\sum_{i=1}^{n} x_i x_i^T + \lambda I$. Consequently, the parameter of global Lipschitz smoothness is $L = \frac{1}{4}\lambda_{\max}(\frac{1}{n}\sum_{i=1}^{n} x_i x_i^T) + \lambda$. It is well known that, with a constant step-size less than (or equal to) $\frac{1}{L}$, a convergence is guaranteed by many algorithms.

However, suppose the algorithm starts at a random $w_0$ (or at $\mathbf{0} \in \mathcal{R}^d$), this bound can be very tight. With more progress being made on approaching an optimal solution (or reducing the training error), it is likely that, for many training samples, $-y_i x_i^T w_t \ll 0$. An immediate implication is that $s_i(w_t)$ defined in (3) becomes smaller and hence the local curvature will be smaller as well. It suggests that, although a large initial step-size could lead to divergence, with more progress made by the algorithm, the parameter of local Lipschitz smoothness tends to be smaller and a larger step-size can be used. That being said, such a dynamic step-size cannot be well defined in the beginning, and a **fully adaptive approach needs to be developed.**

For illustration, we present the inspiring results of an experiment on *real-sim* dataset[1] with $\ell^2$-regularized logistic regression. Figures 1(a) and 1(b) compare the performance of classical *SARAH* with *AI-SARAH* in terms of the evolution of the optimality gap and the squared norm of recursive gradient. As is clear from the figure, *AI-SARAH* displays a significantly faster convergence per effective pass[2].

Now, let us discuss why this could happen. The distribution of $s_i$ as shown in Figured 1(d) indicates that: initially, all $s_i$'s are concentrated at $0.25$; the median continues to reduce within a few effective passes on the training samples; eventually, it stabilizes somewhere below $0.05$. Correspondingly, as presented in Figure 1(c), *AI-SARAH* starts with a conservative step-size dominated by the global Lipschitz smoothness, i.e., $1/\lambda_{max}(\nabla^2 P(w_0))$ (red dots); however, within 5 effective passes, the moving average (magenta dash) and upper-bound (blue line) of the step-size start surpassing the red dots, and eventually stablize above the conservative step-size.

For classical *SARAH*, we configure the algorithm with different values of the fixed step-size, i.e., $\{2^{-2}, 2^{-1}, ..., 2^4\}$, and notice that $2^5$ leads to a divergence. On the other hand, *AI-SARAH* starts with a small step-size, yet achieves a faster convergence per effective pass with an eventual (moving average) step-size larger than $2^5$.

---

[1]The dataset is available at `https://www.csie.ntu.edu.tw/~cjlin/libsvmtools/datasets/`

[2]The effective pass is defined as a complete pass on the training dataset. Each data sample is selected once per effective pass on average.

## 3 Algorithm

We present *AI-SARAH* in Algorithm 1. This algorithm implicitly computes $\alpha_{t-1}$ at any $t \geq 1$ through approximately solving the sub-problem, i.e., $\min_{\alpha>0} \xi_t(\alpha)$, and estimating the local Lipschitz smoothness of a stochastic function; the upper-bound of step-size makes the algorithm stable, and it is updated with exponential smoothing on harmonic mean (of approximate solutions to the sub-problems), which also keep tracks of the local Lipschitz smoothness of a finite sum function, i.e., $P(w)$.

**This algorithm is fully adaptive and requires no efforts of tuning, and can be implemented easily**. Notice that $\beta$ is treated as a smoothing factor in updating the upper-bound of the step-size, and the default setting is $\beta = 0.999$. There exists one hyper-parameter in Algorithm 1, $\gamma$, which defines the early stopping criterion on Line 8, and the default setting is $\gamma = \frac{1}{32}$. We will show later in this chapter that, the performance of this algorithm is not sensitive to the choices of $\gamma$, and this is true regardless of the problems (i.e., regularized/non-regularized logistic regression and different datasets.)

---

**Algorithm 1** *AI-SARAH*

---

1: **Parameter:** $0 < \gamma < 1$ (default $\frac{1}{32}$), $\beta = 0.999$
2: **Initialize:** $\tilde{w}_0$
3: **Set:** $\alpha_{max} = \infty$
4: **for** k = 1, 2, ... **do**
5:     $w_0 = \tilde{w}_{k-1}$
6:     $v_0 = \nabla P(w_0)$
7:     $t = 1$
8:     **while** $\|v_t\|^2 \geq \gamma \|v_0\|^2$ **do**
9:         Select random mini-batch $S_t$ from $[n]$ uniformly with $|S_t| = b$
10:         $\tilde{\alpha}_{t-1} \approx \arg\min_{\alpha>0} \xi_t(\alpha)$
11:         **if** $k = 0$ and $t = 1$ **then**
12:             $\delta_t^k = \frac{1}{\tilde{\alpha}_{t-1}}$
13:         **else**
14:             $\delta_t^k = \beta \delta_{t-1}^k + (1-\beta)\frac{1}{\tilde{\alpha}_{t-1}}$
15:         **end if**
16:         $\alpha_{max} = \frac{1}{\delta_t^k}$
17:         $\alpha_{t-1} = \min\{\tilde{\alpha}_{t-1}, \alpha_{max}\}$
18:         $w_t = w_{t-1} - \alpha_{t-1}v_{t-1}$
19:         $v_t = \nabla f_{S_t}(w_t) - \nabla f_{S_t}(w_{t-1}) + v_{t-1}$
20:         $t = t + 1$
21:     **end while**
22:     Set $\tilde{w}_k = w_t$.
23: **end for**

---

### 3.1 Estimate Local Lipschitz Smoothness

In the previous chapter, we showed that *AI-SARAH* adapts to local Lipschitz smoothness and yields a faster convergence than classical *SARAH*. Then, the question is how to estimate the parameter of local Lipschitz smoothness in practice.

**Can we use line-search?** The standard approach to estimate local Lipschitz smoothness is to use backtracking line-search. Recall *SARAH*'s update rule, i.e., $w_t = w_{t-1} - \alpha_{t-1}v_{t-1}$, where $v_{t-1}$ is a stochastic recursive gradient. The standard procedure is to apply line-search on function $f_{i_t}(w_{t-1} - \alpha v_{t-1})$. However, the main issue is that $-v_{t-1}$ is not necessarily a descent direction.

***AI-SARAH* sub-problem.** Define the sub-problem (as shown on line 10 of Algorithm 1[3]) as

$$\min_{\alpha>0} \xi_t(\alpha) = \min_{\alpha>0} \|\nabla f_{i_t}(w_{t-1} - \alpha v_{t-1}) - \nabla f_{i_t}(w_{t-1}) + v_{t-1}\|^2, \tag{4}$$

where $t \geq 1$ denotes an inner iteration and $i_t$ indexes a random sample selected at $t$. We argue that, by (approximately) solving (4), we can have good estimate of the parameters of the local Lipschitz

---

[3]For sake of simplicity, we use $f_{i_t}$ instead of $f_{S_t}$.

smoothness.

To illustrate this setting, we denote $L_t^i$ the parameter of local Lipschitz smoothness prescribed by $f_{i_t}$ at $w_{t-1}$. Let us focus on a simple quadratic function $f_{i_t}(w) = \frac{1}{2}(x_{i_t}^T w - y_{i_t})^2$. Let $\tilde{\alpha}$ be the optimal step-size along direction $-v_{t-1}$, i.e. $\tilde{\alpha} = \arg\min_\alpha f_{i_t}(w_{t-1} - \alpha v_{t-1})$. Then, the closed form solution of $\tilde{\alpha}$ can be easily derived as $\tilde{\alpha} = \frac{x_{i_t}^T w_{t-1} - y_{i_t}}{x_{i_t}^T v_{t-1}}$, whose value can be positive, negative, bounded or unbounded.

On the other hand, one can compute the step-size implicitly by solving (4) and obtain $\alpha_{t-1}^i$, i.e., $\alpha_{t-1}^i = \arg\min_\alpha \xi_t(\alpha)$. Then, we have

$$\alpha_{t-1}^i = \frac{1}{x_{i_t}^T x_{i_t}},$$

which is exactly $\frac{1}{L_t^i}$ and recall $L_t^i$ is the parameter of local Lipschitz smoothness of $f_{i_t}$.

**Simply put, as quadratic function has a constant Hessian, solving (4) gives exactly $\frac{1}{L_t^i}$. For general (strongly) convex functions, if $\nabla^2 f_{i_t}(w_{t-1})$, does not change too much locally, we can still have a good estimate of $L_t^i$ by solving (4) approximately.**

Based on a good estimate of $L_t^i$, we can then obtain the estimate of the local Lipschitz smoothness of $P(w_{t-1})$. And, that is

$$\bar{L}_t = \frac{1}{n}\sum_{i=1}^n L_t^i = \frac{1}{n}\sum_{i=1}^n \frac{1}{\alpha_{t-1}^i}.$$

Clearly, if a step-size in the algorithm is selected as $1/\bar{L}_t$, then a harmonic mean of the sequence of the step-size's, computed for various component functions could serve as a good adaptive upper-bound on the step-size computed in the algorithm. More details of intuition for the adaptive upper-bound can be found in Appendix A.2.

### 3.2 Compute Step-size and Upper-bound

On Line 10 of Algorithm 1, the sub-problem is a one-dimensional minimization problem, which can be approximately solved by Newton method. Specifically in Algorithm 1, we compute *one-step Newton* at $\alpha = 0$, and that is

$$\tilde{\alpha}_{t-1} = -\frac{\xi_t'(0)}{|\xi_t''(0)|}. \tag{5}$$

Note that, for convex function in general, (5) gives an approximate solution; for functions in particular forms such as quadratic ones, (5) gives an exact solution.

The procedure prescribed in (5) can be implemented very efficiently, and **it does not require any extra (stochastic) gradient computations if compared with classical *SARAH*.** The only extra cost per iteration is to perform two backward passes, i.e., one pass for $\xi_t'(0)$ and the other for $\xi_t''(0)$; see Appendix A.2 for implementation details.

As shown on Lines 11-16 of Algorithm 1, $\alpha_{max}$ is updated at every inner iteration. Specifically, the algorithm starts without an upper bound (i.e., $\alpha_{max} = \infty$ on Line 3); as $\tilde{\alpha}_{t-1}$ being computed at every $t \geq 1$, we employs the exponential smoothing on the harmonic mean of $\{\tilde{\alpha}_{t-1}\}$ to update the upper-bound. For $k \geq 0$ and $t \geq 1$, we define $\alpha_{max} = \frac{1}{\delta_t^k}$, where

$$\delta_t^k = \begin{cases} \frac{1}{\tilde{\alpha}_{t-1}}, & k = 0, t = 1 \\ \beta\delta_{t-1}^k + (1-\beta)\frac{1}{\tilde{\alpha}_{t-1}}, & otherwise \end{cases}$$

and $0 < \beta < 1$. We default $\beta = 0.999$ in Algorithm 1; see Appendix A.2 for details on the design of the adaptive upper-bound.

### 3.3 Choice of $\gamma$

We perform a sensitivity analysis on different choices of $\gamma$. Figures 2 shows the evolution of the squared norm of full gradient, i.e., $\|\nabla P(w)\|^2$, for logistic regression on binary classification problems; see extended results in Appendix A. It is clear that the performance of $\gamma$'s, where, $\gamma \in \{1/8, 1/16, 1/32, 1/64\}$, is consistent with only marginal improvement by using a smaller value. We default $\gamma = 1/32$ in Algorithm 1.

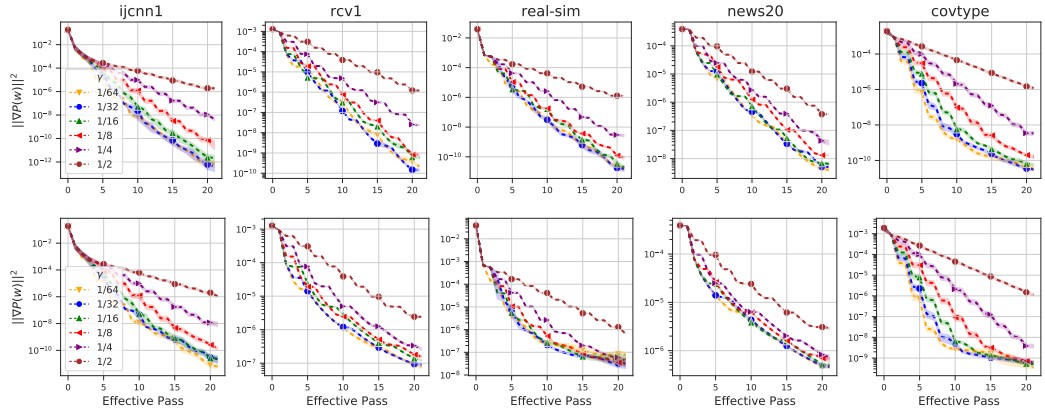

Figure 2: Evolution of $\|\nabla P(w)\|^2$ for $\gamma \in \{\frac{1}{64}, \frac{1}{32}, \frac{1}{16}, \frac{1}{8}, \frac{1}{4}, \frac{1}{2}\}$: regularized (top row) and non-regularized (bottom row) logistic regression on *ijcnn1, rcv1, real-sim, news20* and *covtype*.

### 3.4 CONVERGENCE ANALYSIS

In this section, we provide a convergence analysis of *AI-SARAH* (Algorithm 1) with a i) modified line 10 to $\tilde{\alpha}_{t-1} \approx \arg\min_{\alpha \in [\alpha_{\min}^k, \alpha_{\max}^k]} \xi_t(\alpha)$, and ii) replacing the while loop with a for loop $t \in [m]$, where $\alpha_{\min}^k$ and $\alpha_{\max}^k$ are step-size bounds picked in each outer iteration and $m$ be a hyper-parameter.

For brevity, let us just hint that for many problems (e.g., the one mentioned in Chapter 2), as the solution $w_t$ is approaching $w^*$ the local curvature of $P(w)$ is getting flatter and hence $\alpha_{\max}^k$ could be chosen as a fraction of reciprocal of a smoothness parameter of $f_i(w)$'s over set

$$\mathcal{W}_k := \{w \in \mathcal{R}^d \mid \|\tilde{w}_{k-1} - w\| \leq m \cdot \alpha_{\max}^k \|v_0\|\}, \tag{6}$$

which can be much larger than the fraction of reciprocal of a global smoothness parameter. Let us just remark that for $k$-th outer loop, all iterates $w_t$ will stay inside of $\mathcal{W}_k$ and hence $\alpha_{\max}^k$ is well defined. The parameter $0 < \alpha_{\min}^k \leq \alpha_{\max}^k$ can be chosen arbitrary (e.g., $\alpha_{\min}^k = \alpha_{\max}^k/2$ or even $\alpha_{\min}^k = \alpha_{\max}^k$). We defer the technical details and proofs in Appendix B and C. Let us just state the main convergence theorem here.

**Theorem 3.1.** *Suppose that the functions $f_i(w)$ are convex and smooth with parameter $L_k^{\max}$ over $\mathcal{W}_k$ and $P$ is $\mu$-strongly convex. Let us define*

$$\sigma_m^k = \frac{1}{\mu\alpha_{\min}^k(m+1)} + \frac{\alpha_{\max}^k}{\alpha_{\min}^k} \cdot \frac{\alpha_{\max}^k L_k^{\max}}{2 - \alpha_{\max}^k L_k^{\max}},$$

*and select $m$ and $\alpha_{\max}^k$ such that $\sigma_m^k < 1$, $\forall k \geq 1$. Then, modified Algorithm 1 converges as follows:*

$$\mathbb{E}[\|\nabla P(\tilde{w}_k)\|^2] \leq \left(\prod_{\ell=1}^{k} \sigma_m^\ell\right) \|\nabla P(\tilde{w}_0)\|^2.$$

**Remark:** *SARAH* algorithm is a special case of the modified Algorithm 1 when $\forall k$: $\alpha_{\min}^k = \alpha_{\max}^k = \alpha \leq \frac{1}{2L}$.

## 4 NUMERICAL EXPERIMENT

In this chapter, we present the empirical study on the performance of *AI-SARAH*. For brevity, we present a subset of experiments in the main paper, and defer the full experimental results and implementation details[4] in Appendix A.

The problems we consider in the experiment are $\ell^2$-regularized logistic regression for binary classification problems; see Appendix A for non-regularized case. Given a training sample $(x_i, y_i)$ indexed by $i \in [n]$, the component function $f_i$ is in the form $f_i(w) = \log(1 + \exp(-y_i x_i^T w)) + \frac{\lambda}{2}\|w\|^2$,

---

[4]Code will be made available upon publication.

Table 1: Summary of Datasets from Chang & Lin (2011).

| Dataset | # features | $n$ (# Train) | # Test | % Sparsity |
|---|---|---|---|---|
| *ijcnn1*[1] | 22 | 49,990 | 91,701 | 40.91 |
| *rcv1*[1] | 47,236 | 20,242 | 677,399 | 99.85 |
| *real-sim*[2] | 20,958 | 54,231 | 18,078 | 99.76 |
| *news20*[2] | 1,355,191 | 14,997 | 4,999 | 99.97 |
| *covtype*[2] | 54 | 435,759 | 145,253 | 77.88 |

[1] dataset has default training/testing sanples.
[2] dataset is randomly split by 75%-training & 25%-testing.

where $\lambda = \frac{1}{n}$ for the $\ell^2$-regularized case and $\lambda = 0$ for the non-regularized case.

The datasets chosen for the experiments are *ijcnn1, rcv1, real-sim, news20* and *covtype*. Table 1 shows the basic statistics of the datasets. More details and additional datasets can be found in Appendix A.

We compare *AI-SARAH* with *SARAH*, *SARAH+*, *SVRG* (Johnson & Zhang, 2013), *ADAM* (Kingma & Ba, 2015) and *SGD* with Momentum (Sutskever et al., 2013; Loizou & Richtárik, 2020; 2017). **While *AI-SARAH* does not require hyper-parameter tuning, we fine-tune each of the other algorithms, which yields $\approx 5,000$ runs in total for each dataset and case.**

To be specific, we perform an extensive search on hyper-parameters: (1) *ADAM* and *SGD* with Momentum (*SGD* w/m) are tuned with different values of the (initial) step-size and schedules to reduce the step-size; (2) *SARAH* and *SVRG* are tuned with different values of the (constant) step-size and inner loop size; (3) *SARAH+* is tuned with different values of the (constant) step-size and early stopping parameter. (See Appendix A for detailed tuning plan and the selected hyper-parameters.)

Figure 3 shows the average and total wall clock running time of *AI-SARAH* and the other algorithms. While any individual run of *AI-SARAH* could be 2-5x more time consuming than the other algorithms, its running time is negligible if comparing the total wall clock time. The reason is that *AI-SARAH* does not require any tuning effort, but we have $\approx 5,000$ runs to fine-tune the other algorithms. Figure 4 shows the minimum $\|\nabla P(w)\|^2$ achieved at a few points of effective passes and wall clock time horizon. It is clear that, *AI-SARAH*'s practical speed of convergence is faster than the other algorithms in most cases. Here, we argue that, if given an optimal implementation of *AI-SARAH* (just as that of *ADAM* and other built-in optimizer in Pytorch[5] ), it is likely that our algorithm can be accelerated.

By selecting the fine-tuned hyper-parameters of all other algorithms, we compare them with *AI-SARAH* and show the results in Figures 5-7. For these experiments, we use 10 distinct random seeds to initialize $w$ and generate stochastic mini-batches. And, we use the marked dashes to represent the average and filled areas for 97% confidence intervals.

Figure 5 presents the evolution of $\|\nabla P(w)\|^2$. Obviously from the figure, *AI-SARAH* exhibits the strongest performance in terms of converging to a stationary point: by effective pass, the consistently large gaps are displayed between *AI-SARAH* and the rest; by wall clock time, we notice that *AI-SARAH* achieves the smallest $\|\nabla P(w)\|^2$ at the same time point. This validates our design, that is to leverage local Lipschitz smoothness and achieve a faster convergence than *SARAH* and *SARAH+*.

In terms of minimizing the finite-sum functions, Figure 6 shows that, by effective pass, *AI-SARAH* consistently outperforms *SARAH* and *SARAH+* on all of the datasets with a possible exception on *covtype* dataset. By wall clock time, *AI-SARAH* yields a competitive performance on all of the datasets, and it delivers a stronger performance on *ijcnn1* and *real-sim* than *SARAH*.

For completeness of illustration on the performance, we show the testing accuracy in Figure 7. Clearly, fine-tuned *ADAM* dominates the competition. However, *AI-SARAH* outperforms the other variance reduced methods on most of the datasets from both effective pass and wall clock time perspectives, and achieves the similar levels of accuracy as *ADAM* does on *rcv1*, *real-sim* and *covtype* datasets.

Having illustrated the strong performance of *AI-SARAH*, we continue the presentation by showing the trajectories of the adaptive step-size and upper-bound in Figure 8.

This figure clearly shows that why *AI-SARAH* can achieve such a strong performance, especially on the convergence to a stationary point. As mentioned in previous chapters, the adaptivity is driven by the local Lipschitz smoothness. As shown in Figure 8, *AI-SARAH* starts with conservative step-size and upper-bound, both of which continue to increase while the algorithm progresses towards a stationary point. After a few effective passes, we observe: the step-size and upper-bound are stablized

---

[5]Please see `https://pytorch.org/docs/stable/optim.html` for Pytorch built-in optimizers.

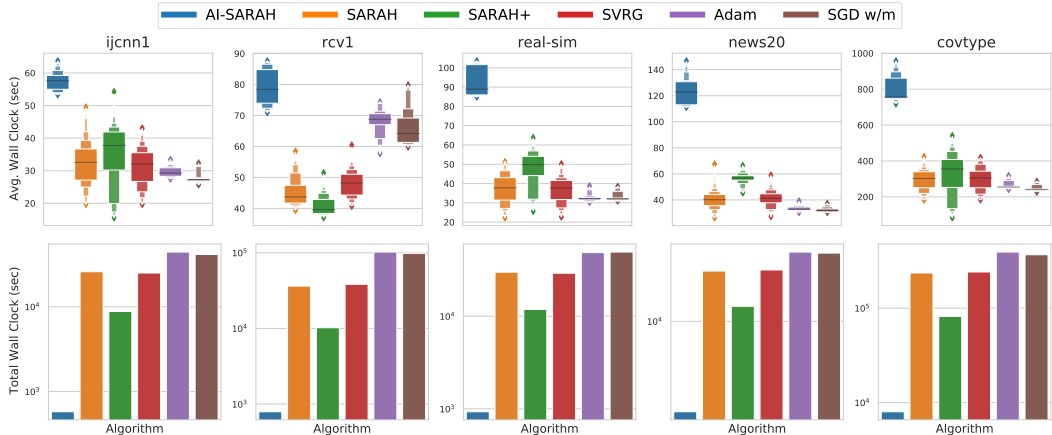

Figure 3: Average (top row) and total (bottom row) running time of *AI-SARAH* and other algorithms for the **regularized** case.

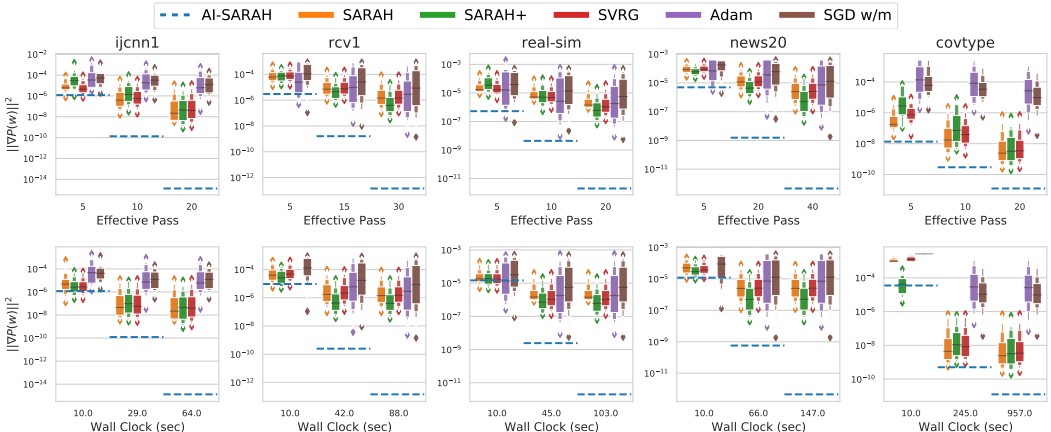

Figure 4: Running minimum per effective pass (top row) and wall clock time (bottom row) of $\|\nabla P(w)\|^2$ between other algorithms with all hyper-parameters configurations and *AI-SARAH* for the **regularized** case. *Note: the horizontal dashes in blue represent the minimum $\|\nabla P(w)\|^2$ achieved by AI-SARAH at certain effective pass or time point.*

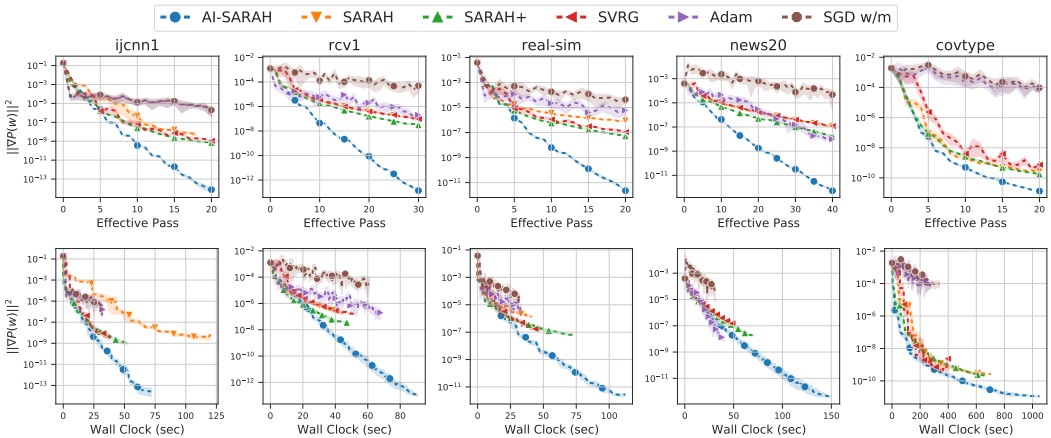

Figure 5: Evolution of $\|\nabla P(w)\|^2$ for the **regularized** case by effective pass (top row) and wall clock time (bottom row).

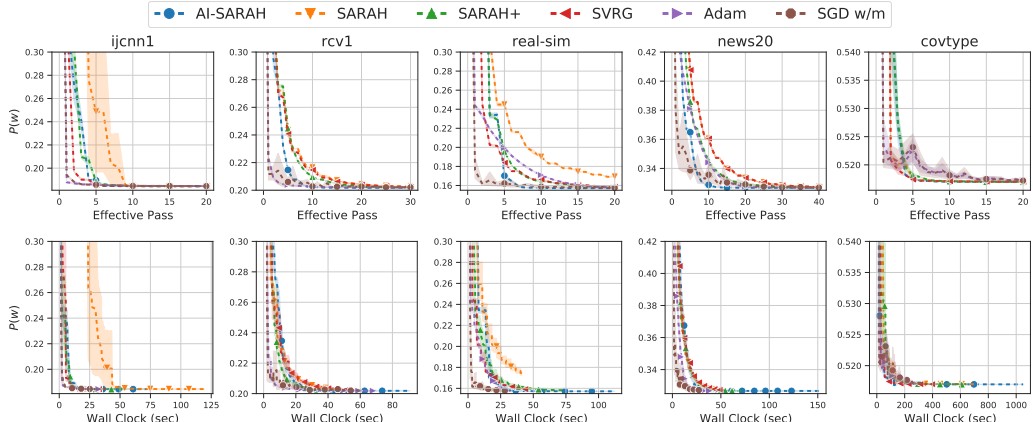

Figure 6: Evolution of $P(w)$ for the **regularized** case by effective pass (top row) and wall clock time (bottom row).

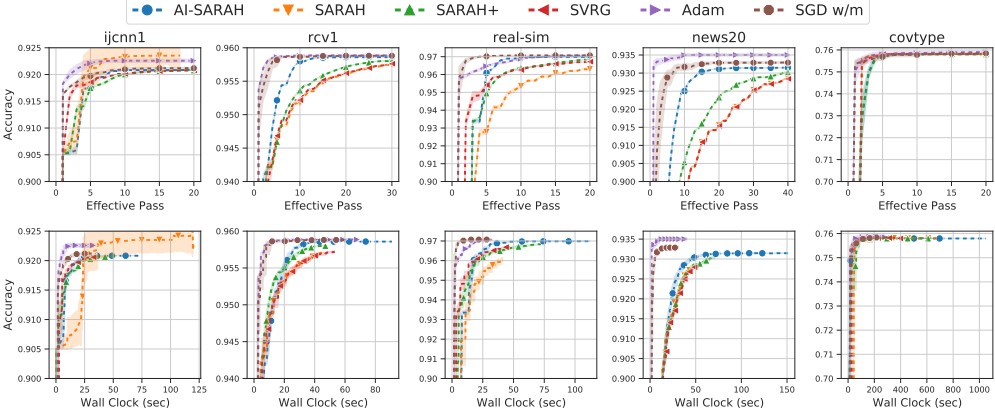

Figure 7: Running maximum of testing accuracy for the **regularized** case by effective pass (top row) and wall clock time (bottom row).

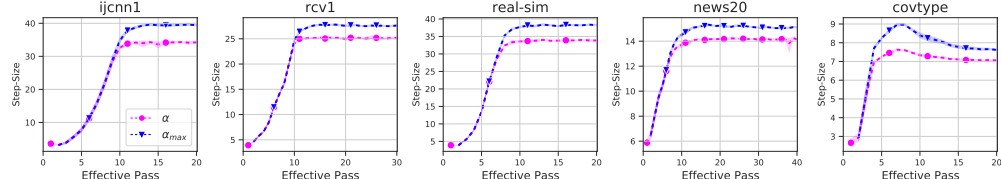

Figure 8: Evolution of *AI-SARAH*'s step-size $\alpha$ and upper-bound $\alpha_{max}$ for the **regularized** case.

due to $\lambda$ (and hence strong convexity). In Appendix A, we can see that, as a result of the function being unregularized, the step-size and upper-bound could be continuously increasing due to the fact that the function is likely non-strongly convex.

## 5 CONCLUSION

In this paper, we propose *AI-SARAH*, a practical variant of stochastic recursive gradient methods. The idea of design is simple yet powerful: by taking advantage of local Lipschitz smoothness, the step-size can be dynamically determined. With intuitive illustration and implementation details, we show how *AI-SARAH* can efficiently estimate local Lipschitz smoothness and how it can be easily implemented in practice. Our algorithm is tune-free and adaptive at full scale. With extensive numerical experiment, we demonstrate that, without (tuning) any hyper-parameters, it delivers a competitive performance compared with *SARAH(+)*, *ADAM* and other first-order methods, all equipped with fine-tuned hyper-parameters.

## ETHICS STATEMENT

This work presents a new algorithm for training machine learning models. We do not foresee any ethical concerns. All datasets used in this work are from the public domain and are commonly used benchmarks in ML papers.

## REPRODUCIBILITY STATEMENT

We uploaded all the codes used to make all the experiments presented in this paper. We have used random seeds to ensure that one can start optimizing the ML models from the same initial starting point as was used in the experiments. We have used only datasets that are in the public domain, and one can download them from the following website `https://www.csie.ntu.edu.tw/~cjlin/libsvmtools/datasets/`. After acceptance, we will include a link to the GitHub repository where we will host the source codes.

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

## APPENDIX

The Appendix is organized as follows. In Chapter A, we present extended details on the design, implementation and results of our numerical experiments. In Chapter B, we present the theoretical analysis of *AI-SARAH*. In Chapter C, we provide the basic definitions, some existing technical preliminaries that are used in our results, and the proofs of the main lemmas and theorems from Chapter B.

## A    EXTENDED DETAILS ON NUMERICAL EXPERIMENT

In this chapter, we present the extended details of the design, implementation and results of the numerical experiments.

### A.1    PROBLEM AND DATA

The machine learning tasks studied in the experiment are binary classification problems. As a common practice in the empirical research of optimization algorithms, the *LIBSVM* datasets[6] are chosen to define the tasks. Specifically, **we selected** 10 **popular binary class datasets: *ijcnn1, rcv1, news20, covtype, real-sim, a1a, gisette, w1a, w8a* and *mushrooms*** (see Table 2 for basic statistics of the datasets).

Table 2: Summary of Datasets.

| Dataset | $d-1$ (# feature) | $n$ (# Train) | $n_{test}$ (# Test) | % Sparsity |
|---|---|---|---|---|
| *ijcnn1*[1] | 22 | 49,990 | 91,701 | 40.91 |
| *rcv1*[1] | 47,236 | 20,242 | 677,399 | 99.85 |
| *news20*[2] | 1,355,191 | 14,997 | 4,999 | 99.97 |
| *covtype*[2] | 54 | 435,759 | 145,253 | 77.88 |
| *real-sim*[2] | 20,958 | 54,231 | 18,078 | 99.76 |
| *a1a*[1] | 123 | 1,605 | 30,956 | 88.73 |
| *gisette*[1] | 5,000 | 6,000 | 1,000 | 0.85 |
| *w1a*[1] | 300 | 2,477 | 47,272 | 96.11 |
| *w8a*[1] | 300 | 49,749 | 14,951 | 96.12 |
| *mushrooms*[2] | 112 | 6,093 | 2,031 | 81.25 |

[1]  dataset has default training/testing samples.
[2]  dataset is randomly split by 75%-training & 25%-testing.

### A.1.1    DATA PRE-PROCESSING

Let $(\chi_i, y_i)$ be a training (or testing) sample indexed by $i \in [n]$ (or $i \in [n_{test}]$), where $\chi_i \in \mathcal{R}^{d-1}$ is a feature vector and $y_i$ is a label. We pre-processed the data such that $\chi_i$ is of a unit length in Euclidean norm and $y_i \in \{-1, +1\}$.

### A.1.2    MODEL AND LOSS FUNCTION

The selected model, $h_i : \mathcal{R}^d \mapsto \mathcal{R}$, is in the linear form

$$h_i(\omega, \varepsilon) = \chi_i^T \omega + \varepsilon, \quad \forall i \in [n], \tag{7}$$

where $\omega \in \mathcal{R}^{d-1}$ is a weight vector and $\varepsilon \in \mathcal{R}$ is a bias term.

For simplicity of notation, from now on, we let $x_i \stackrel{\text{def}}{=} [\chi_i^T \ 1]^T \in \mathcal{R}^d$ be an augmented feature vector, $w \stackrel{\text{def}}{=} [\omega^T \ \varepsilon]^T \in \mathcal{R}^d$ be a parameter vector, and $h_i(w) = x_i^T w$ for $i \in [n]$.

[6]*LIBSVM* datasets are available at `https://www.csie.ntu.edu.tw/~cjlin/libsvmtools/datasets/`.

Given a training sample indexed by $i \in [n]$, the loss function is defined as a logistic regression

$$f_i(w) = \log(1 + \exp(-y_i h_i(w)) + \frac{\lambda}{2}\|w\|^2. \tag{8}$$

In (8), $\frac{\lambda}{2}\|w\|^2$ is the $\ell^2$-regularization of a particular choice of $\lambda > 0$, where we used $\lambda = \frac{1}{n}$ in the experiment; for the non-regularized case, $\lambda$ was set to 0. Accordingly, the finite-sum minimization problem we aimed to solve is defined as

$$\min_{w \in \mathcal{R}^d} \left\{ P(w) \overset{\text{def}}{=} \frac{1}{n} \sum_{i=1}^{n} f_i(w) \right\}. \tag{9}$$

Note that (9) is a convex function. For the $\ell^2$-regularized case, i.e., $\lambda = 1/n$ in (8), (9) is $\mu$-strongly convex and $\mu = \frac{1}{n}$. However, without the $\lambda$, i.e., $\lambda = 0$ in (8), (9) is $\mu$-strongly convex if and only if there there exists $\mu > 0$ such that $\nabla^2 P(w) \succeq \mu I$ for $w \in \mathcal{R}^d$ (provided $\nabla P(w) \in \mathcal{C}$).

## A.2 ALGORITHMS

This section provides the implementation details[7] of the algorithms, practical consideration, and discussions.

### A.2.1 TUNE-FREE *AI-SARAH*

In Chapter 3 of the main paper, we introduced *AI-SARAH*, a tune-free and fully adaptive algorithm. **The implementation of Algorithm 1 was quite straightforward, and we highlight the implementation of Line** 10 **with details**: for logistic regression, the one-dimensional (constrained optimization) sub-problem $\min_{\alpha > 0} \xi_t(\alpha)$ can be approximately solved by computing the Newton step at $\alpha = 0$, i.e., $\tilde{\alpha}_{t-1} = -\frac{\xi'_t(0)}{|\xi''_t(0)|}$. This can be easily implemented with automatic differentiation in Pytorch[8], and only two additional backward passes w.r.t $\alpha$ is needed. For function in some particular form, such as a linear least square loss function, an exact solution in closed form can be easily derived.

As mentioned in Chapter 3, we have an adaptive upper-bound, i.e., $\alpha_{max}$, in the algorithm. To be specific, the algorithm starts without an upper-bound, i.e., $\alpha_{max} = \infty$ on Line 3 of Algorithm 1. Then, $\alpha_{max}$ is updated per (inner) iteration. Recall in Chapter 3, $\alpha_{max}$ is computed as a harmonic mean of the sequence, i.e., $\{\tilde{\alpha}_{t-1}\}$, and an exponential smoothing is applied on top of the simple harmonic mean.

**Having an upper-bound stabilizes the algorithm from stochastic optimization perspective**. For example, when the training error of the randomly selected mini-batch at $w_t$ is drastically reduced or approaching zero, the one-step Newton solution in (5) could be very large, i.e. $\tilde{\alpha}_{t-1} \gg 0$, which could be too aggressive to other mini-batch and hence Problem (1) prescribed by the batch. **On the other hand, making the upper-bound adaptive allows the algorithm to adapt to the local geometry and avoid restrictions** on using a large step-size when the algorithm tries to make aggressive progress with respect to Problem (1). With the adaptive upper-bound being derived by an **exponential smoothing of the harmonic mean, the step-size is determined by emphasizing the current estimate of local geometry while taking into account the history of the estimates**. The exponential smoothing further stabilizes the algorithm by balancing the trade-off of being locally focused (with respect to $f_{S_t}$) and globally focused (with respect to $P$).

It is worthwhile to mention that **Algorithm 1 does not require computing extra gradient of** $f_{S_t}$ **with respect to** $w$ **if compared with** *SARAH* **and** *SARAH+*. At each inner iteration, $t \geq 1$, Algorithm 1 computes $\nabla f_{S_t}(w_{t-1} - \alpha v_{t-1})$ with $\alpha = 0$ just as *SARAH* and *SARAH+* would compute $\nabla f_{S_t}(w_{t-1})$, and the only difference is that $\alpha$ is specified as a variable in Pytorch. After the adaptive step-size $\alpha_{t-1}$ is determined (Line 17), Algorithm 1 computes $\nabla f_{S_t}(w_{t-1} - \alpha_{t-1}v_{t-1})$ just as *SARAH* and *SARAH+* would compute $\nabla f_{S_t}(w_t)$.

In Chapter 3 of the main paper, we discussed the sensitivity of Algorithm 1 on the choice of $\gamma$. Here, we present the full results (on 10 chosen datasets for both $\ell^2$-regularized and non-regularized

---

[7]Code will be made available upon publication.

[8]For detailed description of the automatic differentiation engine in Pytorch, please see `https://pytorch.org/tutorials/beginner/blitz/autograd_tutorial.html`.

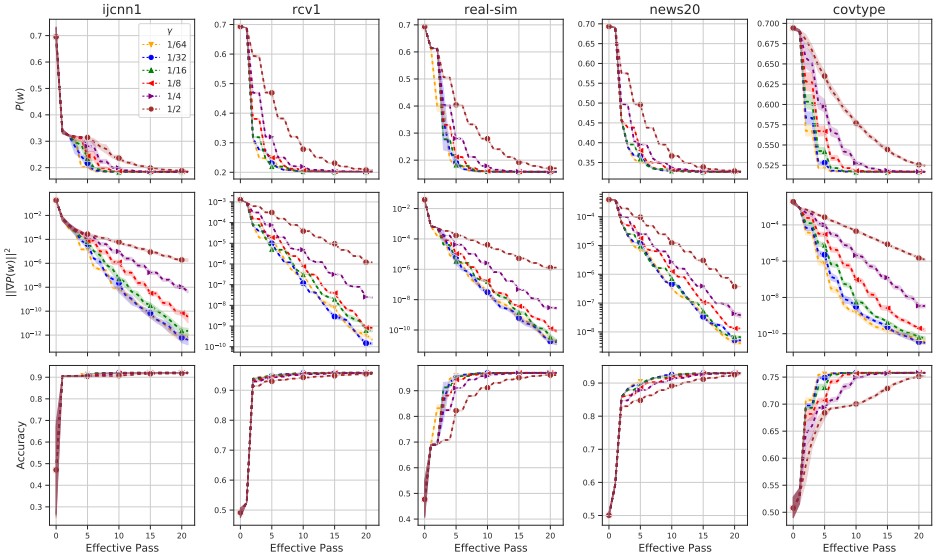

Figure 9: $\ell^2$-regularized case *ijcnn1, rcv1, real-sim, news20* and *covtype* with $\gamma \in \{\frac{1}{64}, \frac{1}{32}, \frac{1}{16}, \frac{1}{8}, \frac{1}{4}, \frac{1}{2}\}$: evolution of $P(w)$ (top row) and $\|\nabla P(w)\|^2$ (middle row) and running maximum of testing accuracy (bottom row).

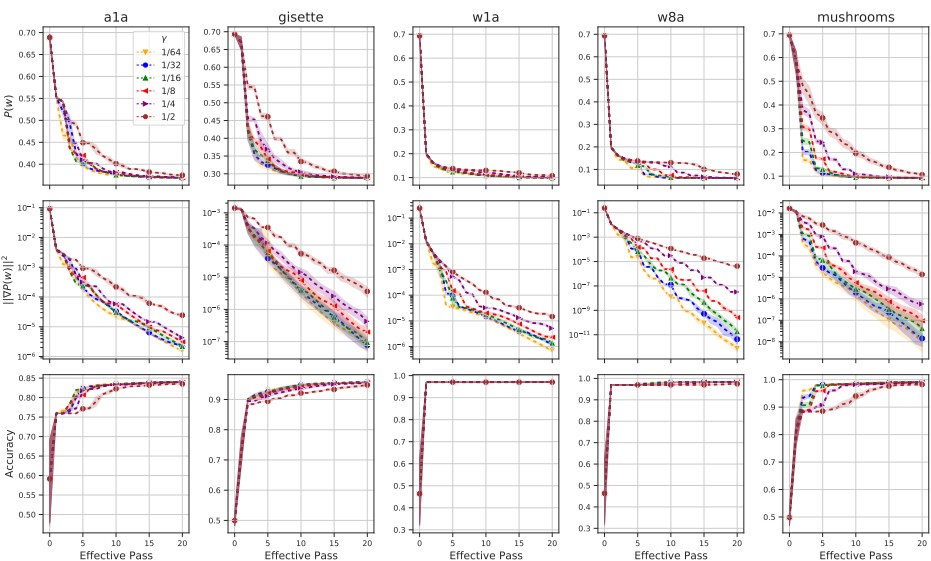

Figure 10: $\ell^2$-regularized case of *a1a, gisette, w1a, w8a* and *mushrooms* with $\gamma \in \{\frac{1}{64}, \frac{1}{32}, \frac{1}{16}, \frac{1}{8}, \frac{1}{4}, \frac{1}{2}\}$: evolution of $P(w)$ (top row) and $\|\nabla P(w)\|^2$ (middle row) and running maximum of testing accuracy (bottom row).

cases) in Figures 9, 10, 11, and 12. Note that, in this experiment, we chose $\gamma \in \{\frac{1}{64}, \frac{1}{32}, \frac{1}{16}, \frac{1}{8}, \frac{1}{4}, \frac{1}{2}\}$, and for each $\gamma$, dataset and case, we used 10 distinct random seeds and ran each experiment for 20 effective passes.

### A.2.2 OTHER ALGORITHMS

In our numerical experiment, we compared the performance of **TUNE-FREE** *AI-SARAH* (Algorithm 1) with that of 5 **FINE-TUNED** state-of-the-art (stochastic variance reduced or adaptive) first-order

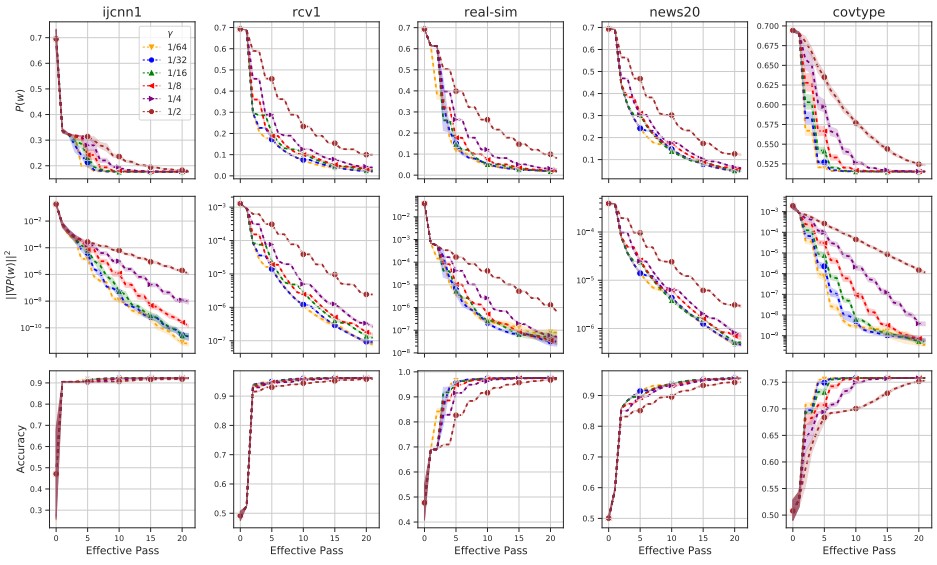

Figure 11: Non-regularized case *ijcnn1, rcv1, real-sim, news20* and *covtype* with $\gamma \in \{\frac{1}{64}, \frac{1}{32}, \frac{1}{16}, \frac{1}{8}, \frac{1}{4}, \frac{1}{2}\}$: evolution of $P(w)$ (top row) and $\|\nabla P(w)\|^2$ (middle row) and running maximum of testing accuracy (bottom row).

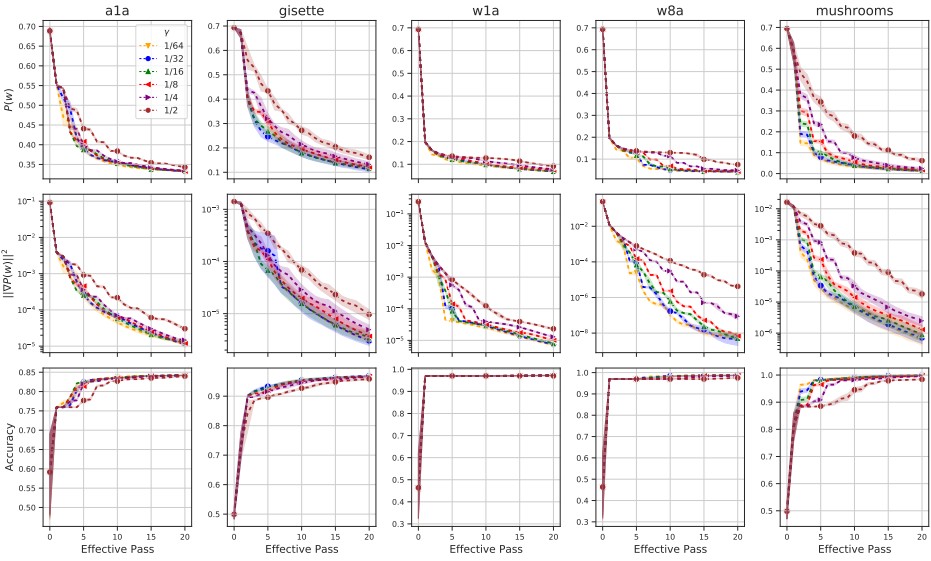

Figure 12: Non-regularized case *a1a, gisette, w1a, w8a* and *mushrooms* with $\gamma \in \{\frac{1}{64}, \frac{1}{32}, \frac{1}{16}, \frac{1}{8}, \frac{1}{4}, \frac{1}{2}\}$: evolution of $P(w)$ (top row) and $\|\nabla P(w)\|^2$ (middle row) and running maximum of testing accuracy (bottom row).

Table 3: Tuning Plan - Choice of Hyper-parameters.

| Method | # Configuration | Step-Size | Schedule (%)[1] | Inner Loop Size (# Effective Pass) | Early Stopping ($\gamma$) |
|---|---|---|---|---|---|
| SARAH | 160 | $\{0.1, 0.2, ..., 1\}/L$ | n/a | $\{0.5, 0.6, ..., 2\}$ | n/a |
| SARAH+ | 50 | $\{0.1, 0.2, ..., 1\}/L$ | n/a | n/a | $1/\{2, 4, 8, 16, 32\}$ |
| SVRG | 160 | $\{0.1, 0.2, ..., 1\}/L$ | n/a | $\{0.5, 0.6, ..., 2\}$ | n/a |
| ADAM[2] | 300 | $[10^{-3}, 10]$ | $\{0, 1, 5, 10, 15\}$ | n/a | n/a |
| SGD w/m[3] | 300 | $[10^{-3}, 10]$ | $\{0, 1, 5, 10, 15\}$ | n/a | n/a |

[1] Step-size is scheduled to decrease by $X\%$ every effective pass over the training samples.
[2] $\beta_1 = 0.9, \beta_2 = 0.999$.
[3] $\beta = 0.9$.

Table 4: Running Budget (# Effective Pass).

| Dataset | Regularized | Non-regularized |
|---|---|---|
| ijcnn1 | 20 | 20 |
| rcv1 | 30 | 40 |
| news20 | 40 | 50 |
| covtype | 20 | 20 |
| real-sim | 20 | 30 |
| a1a | 30 | 40 |
| gisette | 30 | 40 |
| w1a | 40 | 50 |
| w8a | 30 | 40 |
| mushrooms | 30 | 40 |

methods: *SARAH*, *SARAH+*, *SVRG*, *ADAM* and *SGD* with Momentum (*SGD w/m*). These algorithms were implemented in Pytorch, where *ADAM* and *SGD* w/m are built-in optimizers of Pytorch.

**Hyper-parameter tuning.** For *ADAM* and *SGD* w/m, we selected 60 different values of the (initial) step-size on the interval $[10^{-3}, 10]$ and 5 different schedules to decrease the step-size after every effective pass on the training samples; for *SARAH* and *SVRG*, we selected 10 different values of the (constant) step-size and 16 different values of the inner loop size; for *SARAH+*, the values of step-size were selected in the same way as that of *SARAH* and *SVRG*. In addition, we chose 5 different values of the inner loop early stopping parameter. Table 3 presents the detailed tuning plan for these algorithms.

*Selection criteria:*

We defined the best hyper-parameters as the ones yielding the minimum ending value of the loss function, where the running budget is presented in Table 4. Specifically, the criteria are: (1) filtering out the ones exhibited a "spike" of the loss function, i.e., the initial value of the loss function is surpassed at any point within the budget; (2) selecting the ones achieved the minimum ending value of the loss function.

*Hightlights of the hyper-parameter search:*

- To take into account the randomness in the performance of these algorithms provided different hyper-parameters, we ran each configuration with 5 distinct random seeds. **The total number of runs for each dataset and case is** $4,850$.
- Tables 5 and 6 present the best hyper-parameters selected from the candidates for the regularized and non-regularized cases.
- Figures 13, 14, 15 and 16 show the performance of different hyper-parameters for all tuned algorithms; it is clearly that, **the performance is highly dependent on the choices of hyper-parameter for *SARAH*, *SARAH+*, and *SVRG***. And, **the performance of *ADAM* and *SGD* w/m are very SENSITIVE to the choices of hyper-parameter**.

**Global Lipschitz smoothness of** $P(w)$**.** Tuning the (constant) step-size of *SARAH*, *SARAH+* and *SVRG* requires the parameter of (global) Lipschitz smoothness of $P(w)$, denoted the (global) Lipschitz

Table 5: Fine-tuned Hyper-parameters - $\ell^2$-regularized Case.

| Dataset | ADAM $(\alpha_0, x\%)$ | SGD w/m $(\alpha_0, x\%)$ | SARAH $(\alpha, m)$ | SARAH+ $(\alpha, \gamma)$ | SVRG $(\alpha, m)$ |
|---|---|---|---|---|---|
| ijcnn1 | (0.07, 15%) | (0.4, 15%) | (3.153, 1015) | (3.503, 1/32) | (3.503, 1562) |
| rcv1 | (0.016, 10%) | (4.857, 10%) | (3.924, 600) | (3.924, 1/32) | (3.924, 632) |
| news20 | (0.028, 15%) | (6.142, 10%) | (3.786, 468) | (3.786, 1/32) | (3.786, 468) |
| covtype | (0.07, 15%) | (0.4, 15%) | (2.447, 13616) | (2.447, 1/32) | (2.447, 13616) |
| real-sim | (0.16, 15%) | (7.428, 15%) | (3.165, 762) | (3.957, 1/32) | (3.957, 1694) |
| a1a | (0.7, 15%) | (4.214, 15%) | (2.758, 50) | (2.758, 1/32) | (2.758, 50) |
| gisette | (0.028, 15%) | (8.714, 10%) | (2.320, 186) | (2.320, 1/16) | (2.320, 186) |
| w1a | (0.1, 10%) | (3.571, 10%) | (3.646, 60) | (3.646, 1/32) | (3.646, 76) |
| w8a | (0.034, 15%) | (2.285, 15%) | (2.187, 543) | (3.645, 1/32) | (3.645, 1554) |
| mushrooms | (0.220, 15%) | (3.571, 0%) | (2.682, 190) | (2.682, 1/32) | (2.682, 190) |

Table 6: Fine-tuned Hyper-parameters - Non-regularized Case.

| Dataset | ADAM $(\alpha_0, x\%)$ | SGD w/m $(\alpha_0, x\%)$ | SARAH $(\alpha, m)$ | SARAH+ $(\alpha, \gamma)$ | SVRG $(\alpha, m)$ |
|---|---|---|---|---|---|
| ijcnn1 | (0.1, 15%) | (0.58, 15%) | (3.153, 1015) | (3.503, 1/32) | (3.503, 1562) |
| rcv1 | (5.5, 10%) | (10.0, 0%) | (3.925, 632) | (3.925, 1/32) | (3.925, 632) |
| news20 | (1.642, 10%) | (10.0, 0%) | (3.787, 468) | (3.787, 1/32) | (3.787, 468) |
| covtype | (0.16, 15%) | (2.2857, 15%) | (2.447, 13616) | (2.447, 1/32) | (2.447, 13616) |
| real-sim | (2.928, 15%) | (10.0, 0%) | (3.957, 1609) | (3.957, 1/16) | (3.957, 1694) |
| a1a | (1.642, 15%) | (6.785, 1%) | (2.763, 50) | (2.763, 1/32) | (2.763, 50) |
| gisette | (2.285, 1%) | (10.0, 0%) | (2.321, 186) | (2.321, 1/32) | (2.321, 186) |
| w1a | (8.714, 10%) | (10.0, 0%) | (3.652, 76) | (3.652, 1/32) | (3.652, 76) |
| w8a | (0.16, 10%) | (10.0, 5%) | (2.552, 543) | (3.645, 1/32) | (3.645, 1554) |
| mushrooms | (10.0, 0%) | (10.0, 0%) | (2.683, 190) | (2.683, 1/32) | (2.683, 190) |

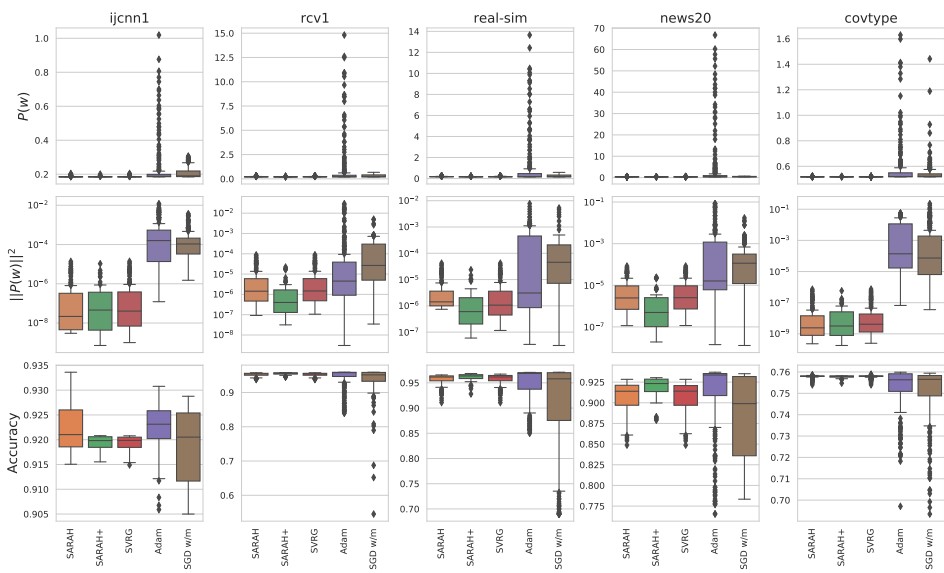

Figure 13: Ending loss (top row), ending squared norm of full gradient (middle row), maximum testing accuracy (bottom row) of different hyper-paramters and algorithms for the $\ell^2$-**regularized case** on *ijcnn1, rcv1, real-sim, news20* and *covtype* datasets.

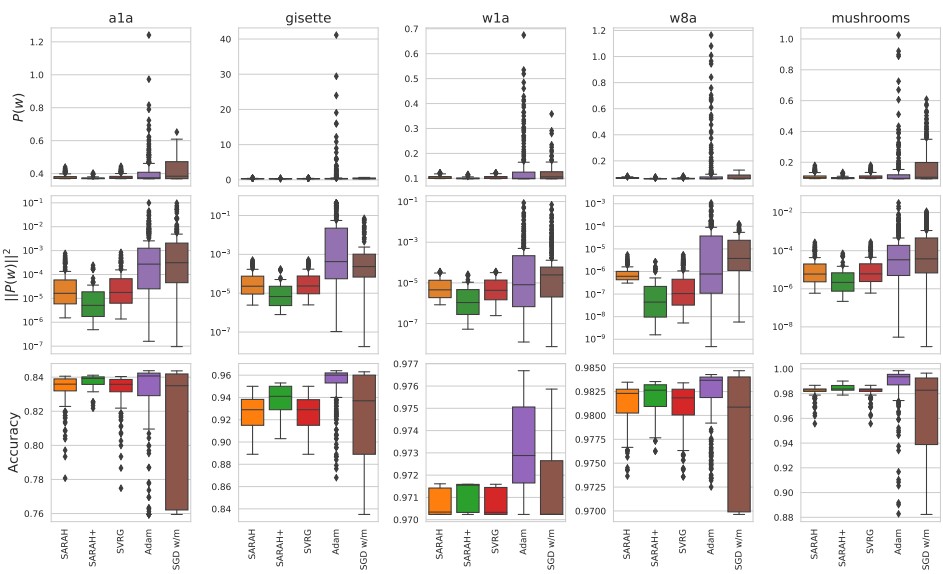

Figure 14: Ending loss (top row), ending squared norm of full gradient (middle row), maximum testing accuracy (bottom row) of different hyper-paramters and algorithms for the $\ell^2$-**regularized case** on *a1a, gisette, w1a, w8a* and *mushrooms* datasets.

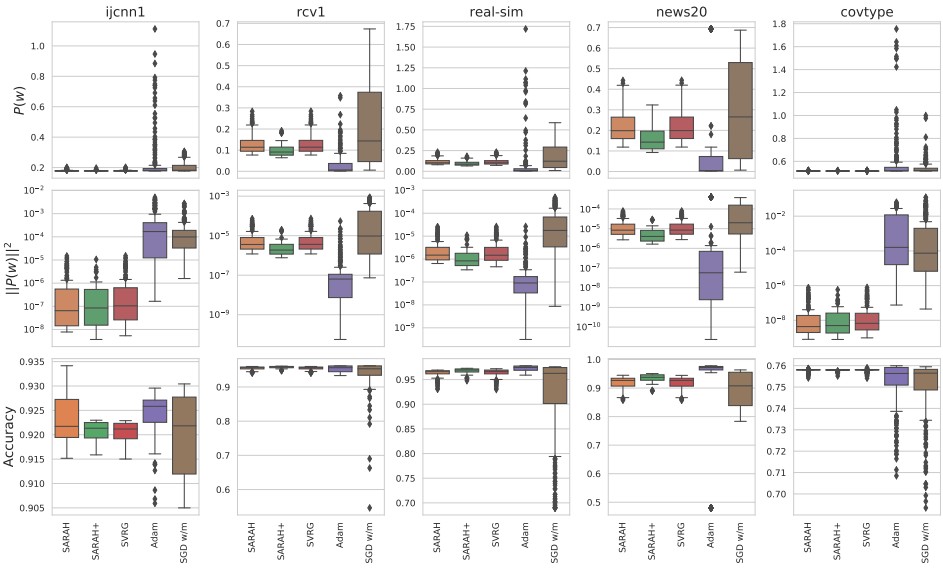

Figure 15: Ending loss (top row), ending squared norm of full gradient (middle row), maximum testing accuracy (bottom row) of different hyper-paramters and algorithms for the **non-regularized case** on *ijcnn1, rcv1, real-sim, news20* and *covtype* datasets.

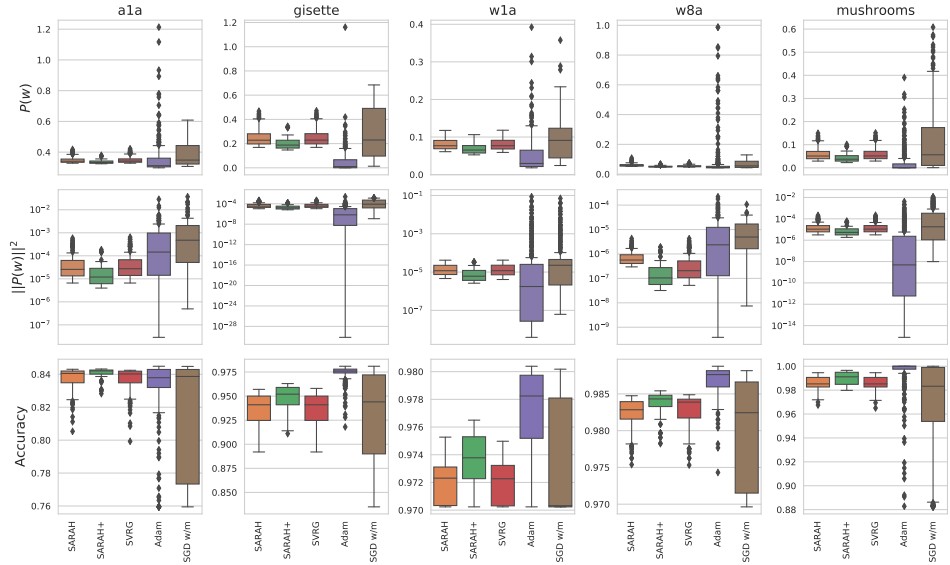

Figure 16: Ending loss (top row), ending squared norm of full gradient (middle row), maximum testing accuracy (bottom row) of different hyper-paramters and algorithms for the **non-regularized case** on *a1a, gisette, w1a, w8a* and *mushrooms* datasets.

Table 7: Global Lipschitz Constant $L$

| Dataset | Regularized | Non-regularized |
|---------|-------------|-----------------|
| *ijcnn1* | 0.285408 | 0.285388 |
| *rcv1* | 0.254812 | 0.254763 |
| *news20* | 0.264119 | 0.264052 |
| *covtype* | 0.408527 | 0.408525 |
| *real-sim* | 0.252693 | 0.252675 |
| *a1a* | 0.362456 | 0.361833 |
| *gisette* | 0.430994 | 0.430827 |
| *w1a* | 0.274215 | 0.273811 |
| *w8a* | 0.274301 | 0.274281 |
| *mushrooms* | 0.372816 | 0.372652 |

constant $L$, and it can be computed as, given (8) and (9),

$$L = \frac{1}{4}\lambda_{max}\left(\frac{1}{n}\sum_{i=1}^{n} x_i x_i^T\right) + \lambda,$$

where $\lambda_{max}(A)$ denotes the largest eigenvalue of $A$ and $\lambda$ is the penalty term of the $\ell^2$-regularization in (8). Table 7 shows the values of $L$ for the regularized and non-regularized cases on the chosen datasets.

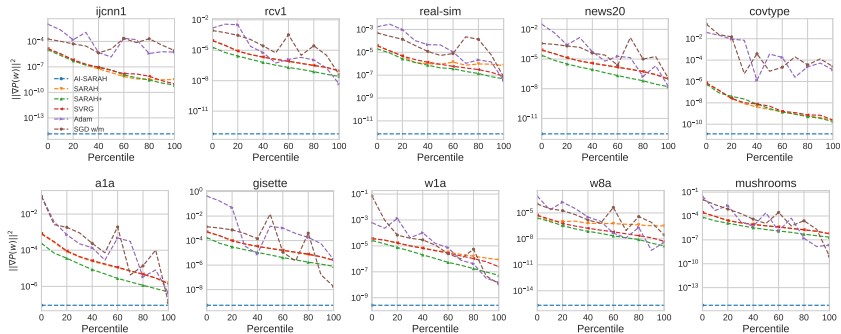

Figure 17: Average ending $\|\nabla P(w)\|^2$ for $\ell^2$-regularized case - *AI-SARAH* vs. Other Algorithms: *AI-SARAH is shown as the horizontal lines; for each of the other algorithms, the average ending* $\|\nabla P(w)\|^2$ *from different configurations of hyper-parameters are indexed from* 0 *percentile (the worst choice) to* 100 *percentile (the best choice); see Section A.2.2 for details of the selection criteria.*

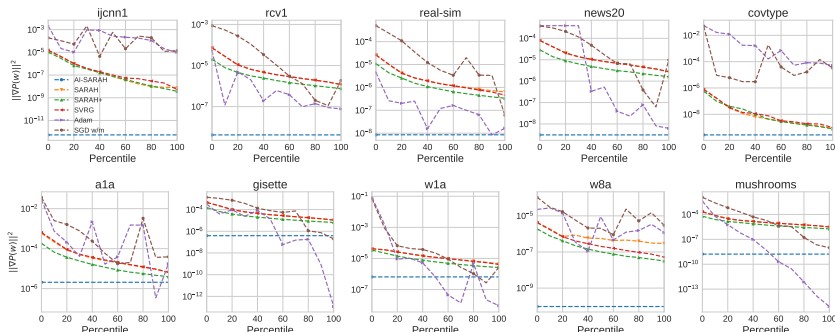

Figure 18: Average ending $\|\nabla P(w)\|^2$ for non-regularized case - *AI-SARAH* vs. Other Algorithms.

## A.3 EXTENDED RESULTS OF EXPERIMENT

In Chapter 4, we compared tune-free & fully adaptive *AI-SARAH* (Algorithm 1) with fine-tuned *SARAH*, *SARAH+*, *SVRG*, *ADAM* and *SGD* w/m. In this section, we present the extended results of our empirical study on the performance of *AI-SARAH*.

Figures 17 and 18 compare the average ending $\|\nabla P(w)\|^2$ achieved by *AI-SARAH* with the other algorithms, configured with all candidate hyper-parameters.

It is clear that,

- without tuning, *AI-SARAH* achieves the best convergence (to a stationary point) in practice on most of the datasets for both cases;
- while fine-tuned *ADAM* achieves a better result for the non-regularized case on *a1a, gisette, w1a* and *mushrooms*, *AI-SARAH* outperforms *ADAM* for at least $80\%$ (*a1a*), $55\%$ (*gisette*), $50\%$ (*w1a*), and $50\%$ (*mushrooms*) of all candidate hyper-parameters.

Figure 19 shows the results of the non-regularized case for *ijcnn1, rcv1, real-sim, news20* and *covtype* datasets. Figures 20 and 21 present the results of the $\ell^2$-regularized case and non-regularized case respectively on *a1a, gisette, w1a, w8a* and *mushrooms* datasets. For completeness of presentation, we present the evolution of *AI-SARAH*'s step-size and upper-bound on *a1a, gisette, w1a, w8a* and *mushrooms* datasets in Figures 22 and 23. Consistent with the results shown in Chapter 4 of the main paper, *AI-SARAH* delivers a competitive performance in practice.

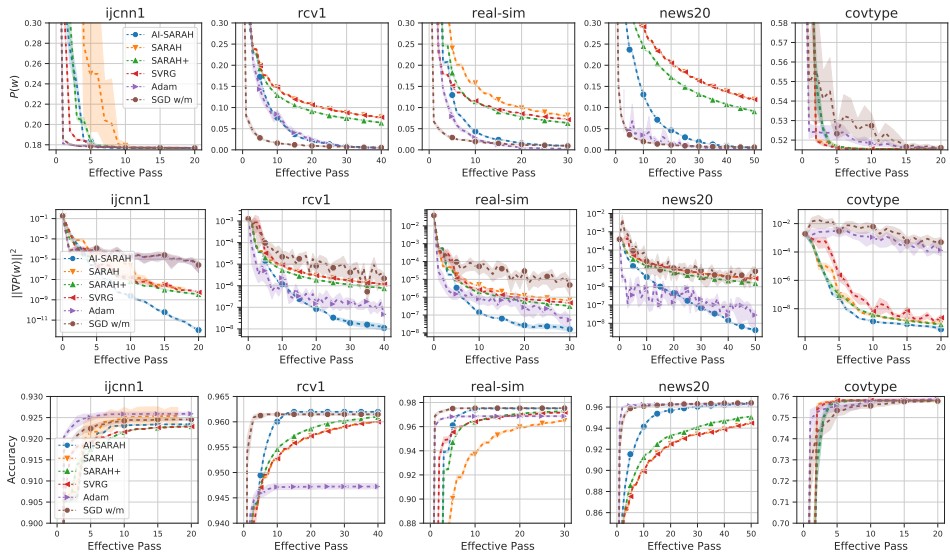

Figure 19: Non-regularized case: evolution of $P(w)$ (top row), $\|\nabla P(w)\|^2$ (middle row), and running maximum of testing accuracy (bottom row).

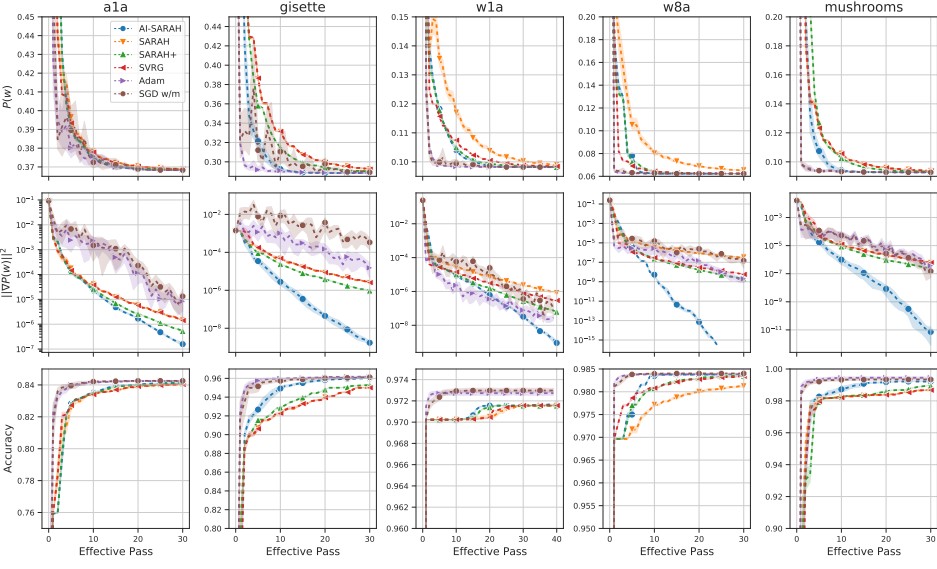

Figure 20: $\ell^2$-regularized case: evolution of $P(w)$ (top row), $\|\nabla P(w)\|^2$ (middle row), and running maximum of testing accuracy (bottom row).

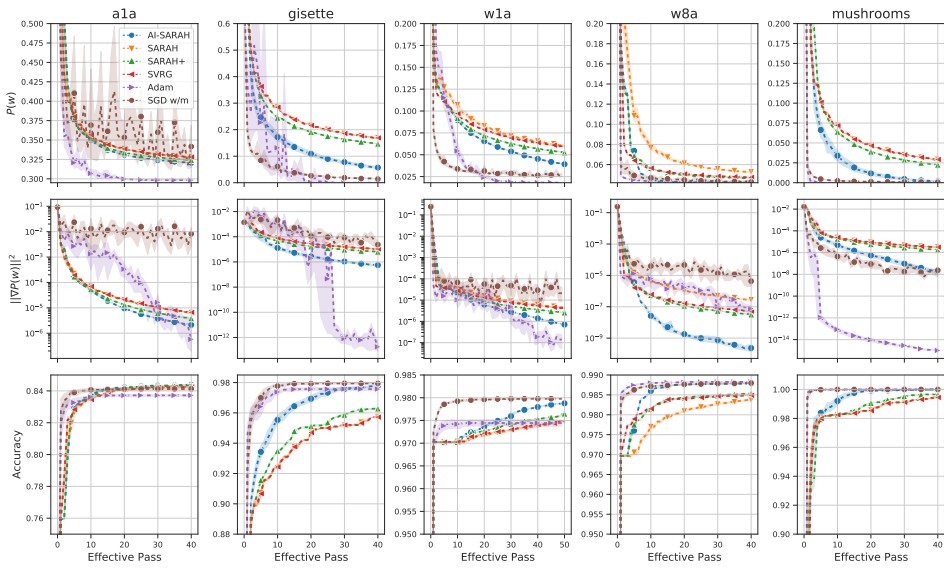

Figure 21: Non-regularized case: evolution of $P(w)$ (top row), $\|\nabla P(w)\|^2$ (middle row), and running maximum of testing accuracy (bottom row).

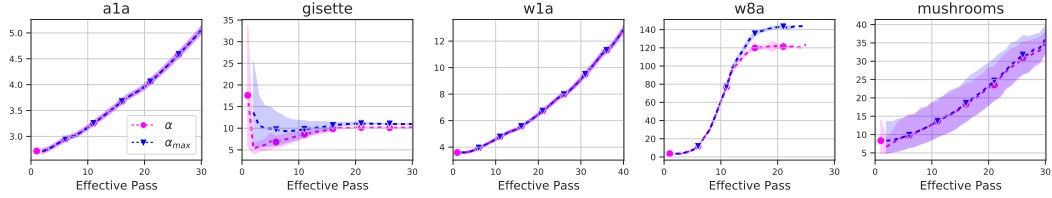

Figure 22: $\ell^2$-regularized case: evolution of *AI-SARAH*'s step-size $\alpha$ and upper-bound $\alpha_{max}$.

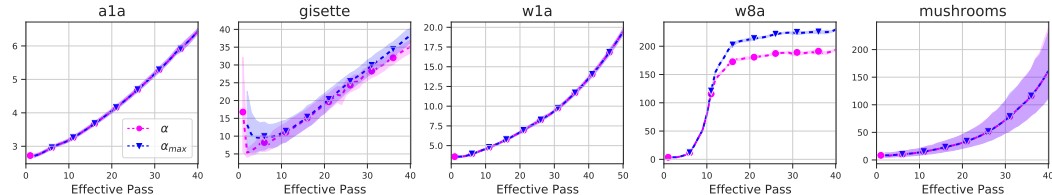

Figure 23: Non-regularized case: evolution of *AI-SARAH*'s step-size $\alpha$ and upper-bound $\alpha_{max}$.

## B   THEORETICAL ANALYSIS

In this chapter, we provide a convergence analysis of *AI-SARAH* (Algorithm 1) with a i) modified line 10 to $\tilde{\alpha}_{t-1} \approx \arg\min_{\alpha \in [\alpha_{\min}^k, \alpha_{\max}^k]} \xi_t(\alpha)$, and ii) replacing the while loop with a for loop $t \in [m]$, where $\alpha_{\min}^k$ and $\alpha_{\max}^k$ are step-size bounds picked in each outer iteration and $m$ be a hyper-parameter.

### B.1   *AI-SARAH* AND *SARAH*

Like *SVRG* and *SARAH*, *AI-SARAH*'s loop structure of this algorithm is divided into the outer loop, where a full gradient is computed, and the inner loop, where only stochastic gradient is computed. However, unlike *SVRG* and *SARAH*, the step-size is computed implicitly. In particular, at each iteration $t \in [m]$ of the inner loop, the step-size is chosen by approximately solving a simple one-dimensional constrained optimization problem. Define the (modified) sub-problem (optimization problem) at $t \geq 1$ as

$$\min_{\alpha \in [\alpha_{\min}^k, \alpha_{\max}^k]} \xi_t(\alpha), \tag{10}$$

where $\xi_t(\alpha) := \|v_t\|^2 = \|\nabla f_{S_t}(w_{t-1} - \alpha v_{t-1}) - \nabla f_{S_t}(w_{t-1}) + v_{t-1}\|^2$, $\alpha_{\min}^k$ and $\alpha_{\max}^k$ are lower-bound and upper-bound of the step-size respectively. These bounds do not allow large fluctuations of the (adaptive) step-size. We denote $\alpha_{t-1}$ the approximate solution of (10). Now, let us present some remarks regarding *AI-SARAH*.

**Remark B.1.** *As we will explain with more details in the following subsections, the values of $\alpha_{\min}^k$ and $\alpha_{\max}^k$ cannot be arbitrarily large. To guarantee convergence, we will need to assume that $\alpha_{\max}^k \leq \frac{2}{L_k^{\max}}$, where $L_k^{\max} = \max_{i \in [n]} L_k^i$. Here, $L_k^i$ is the local smoothness parameter of $f_i$ defined on a working-set for each outer loop (see Definition B.6).*

**Remark B.2.** *SARAH Nguyen et al. (2017) can be seen as a special case of AI-SARAH, where $\alpha_{\min}^k = \alpha_{\max}^k = \alpha$ for all outer loops ($k \geq 1$). In other words, a constant step-size is chosen for the algorithm. However, if $\alpha_{\min}^k < \alpha_{\max}^k$, then the selection of the step-size in AI-SARAH allows a faster convergence of $\|v_t\|^2$ than SARAH in each inner loop.*

**Remark B.3.** *At $t \geq 1$, let us select a mini-batch of size $n$, i.e., $|S_t| = n$. In this case, AI-SARAH is equivalent to deterministic gradient descent with a very particular way of selecting the step-size, i.e. by solving the following problem*

$$\min_{\alpha \in [\alpha_{\min}^k, \alpha_{\max}^k]} \xi_t(\alpha),$$

*where $\xi_t(\alpha) = \|\nabla P(w_{t-1} - \alpha \nabla P(w_{t-1}))\|^2$. In other words, the step-size is selected to minimize the squared norm of the full gradient with respect to $w_t$.*

### B.2   DEFINITIONS / ASSUMPTIONS

First, we present the main definitions and assumptions that are used in our convergence analysis.

**Definition B.4.** *Function $f : \mathcal{R}^d \to \mathcal{R}$ is L-smooth if: $f(x) \leq f(y) + \langle \nabla f(y), x - y \rangle + \frac{L}{2}\|x - y\|^2, \forall x, y \in \mathcal{R}^d$,*
*and it is $L_C$-smooth if:*

$$f(x) \leq f(y) + \langle \nabla f(y), x - y \rangle + \frac{L_C}{2}\|x - y\|^2, \forall x, y \in \mathcal{C}.$$

**Definition B.5.** *Function $f : \mathcal{R}^d \to \mathcal{R}$ is $\mu$-strongly convex if: $f(x) \geq f(y) + \langle \nabla f(y), x - y \rangle + \frac{\mu}{2}\|x - y\|^2, \forall x, y \in \mathcal{R}^d$. If $\mu = 0$ then function $f$ is a (non-strongly) convex function.*

Having presented the two main definitions for the class of problems that we are interested in, let us now present the working-set $\mathcal{W}_k$ which contains all iterates produced in the $k$-th outer loop of (the modified) Algorithm 1.

**Definition B.6** (Working-Set $\mathcal{W}_k$)**.** *For any outer loop $k \geq 1$ in (the modified) Algorithm 1, starting at $\tilde{w}_{k-1}$ we define*

$$\mathcal{W}_k := \{w \in \mathcal{R}^d \mid \|\tilde{w}_{k-1} - w\| \leq m \cdot \alpha_{\max}^k \|v_0\|\}. \tag{11}$$

Note that the working-set $\mathcal{W}_k$ can be seen as a ball of all vectors $w$'s, which are not further away from $\tilde{w}_{k-1}$ than $m \cdot \alpha_{\max}^k \|v_0\|$. Here, recall that $m$ is the total number of iterations of an inner loop, $\alpha_{\max}^k$ is an upper bound of the step-size $\alpha_{t-1}$, $\forall t \in [m]$, and $\|v_0\|$ is simply the norm of the full gradient evaluated at the starting point $\tilde{w}_{k-1}$ in the outer loop.

By combining Definition B.4 with the working-set $\mathcal{W}_k$, we are now ready to provide the main assumption used in our analysis.

**Assumption 1.** *Functions $f_i$, $i \in [n]$, of problem (1) are $L_{\mathcal{W}_k}^i$-smooth. Since we only focus on the working-set $\mathcal{W}_k$, we simply write $L_k^i$-smooth.*

Let us denote $L_i$ the smoothness parameter of function $f_i$, $i \in [n]$, in the domain $\mathcal{R}^d$. Then, it is easy to see that $L_k^i \leq L_i$, $\forall i \in [n]$. In addition, under Assumption 1, it holds that function $P$ is $\bar{L}_k$-smooth in the working-set $\mathcal{W}_k$, where $\bar{L}_k = \frac{1}{n} \sum_{i=1}^n L_k^i$.

As we will explain with more details in the next section for our theoretical results, we will assume that $\alpha_{\max}^k \leq \frac{2}{L_k^{\max}}$, where $L_k^{\max} = \max_{i \in [n]} L_k^i$.

### B.3 CONVERGENCE GUARANTEES

Now, we can derive the convergence rate of *AI-SARAH*. Here, we highlight that, all of our theoretical results can be applied to *SARAH*. We also note that, some quantities involved in our results, such as $L_k$ and $L_k^{\max}$, are dependent upon the working set $\mathcal{W}_k$ (defined for each outer loop $k \geq 1$). Similar to Nguyen et al. (2017), we start by presenting two important lemmas, serving as the foundation of our theory.

The first lemma provides an upper bound on the quantity $\sum_{t=0}^m \mathbb{E}[\|\nabla P(w_t)\|^2]$. Note that it does not require any convexity assumption.

**Lemma B.7.** *Fix a outer loop $k \geq 1$ and consider Algorithm 1 with $\alpha_{\max}^k \leq 1/\bar{L}_k$. Under Assumption 1, $\sum_{t=0}^m \mathbb{E}[\|\nabla P(w_t)\|^2] \leq \frac{2}{\alpha_{\min}^k}\mathbb{E}[P(w_0) - P(w^*)] + \frac{\alpha_{\max}^k}{\alpha_{\min}^k}\sum_{t=0}^m \mathbb{E}[\|\nabla P(w_t) - v_t\|^2]$.*

The second lemma provides an informative bound on the quantity $\mathbb{E}[\|\nabla P(w_t) - v_t\|^2]$. Note that it requires convexity of component functions $f_i$, $i \in [n]$.

**Lemma B.8.** *Fix a outer loop $k \geq 1$ and consider Algorithm 1 with $\alpha_{\max}^k < 2/L_k^{\max}$. Suppose $f_i$ is convex for all $i \in [n]$. Then, under Assumption 1, for any $t \geq 1$ :*

$$\mathbb{E}[\|\nabla P(w_t) - v_t\|^2] \leq \left( \frac{\alpha_{\max}^k L_k^{\max}}{2 - \alpha_{\max}^k L_k^{\max}} \right) \mathbb{E}[\|v_0\|^2].$$

Equipped with the above lemmas, we can then present our main theorem and show the linear convergence of (the modified) Algorithm 1 for solving strongly convex smooth problems.

**Theorem B.9.** *Suppose that Assumption 1 holds and $P$ is strongly convex with convex component functions $f_i$, $i \in [n]$. Let us define*

$$\sigma_m^k = \frac{1}{\mu\alpha_{\min}^k(m+1)} + \frac{\alpha_{\max}^k}{\alpha_{\min}^k} \cdot \frac{\alpha_{\max}^k L_k^{\max}}{2 - \alpha_{\max}^k L_k^{\max}},$$

*and select $m$ and $\alpha_{\max}^k$ such that $\sigma_m^k < 1$, $\forall k \geq 1$. Then, Algorithm 1 converges as follows:*

$$\mathbb{E}[\|\nabla P(\tilde{w}_k)\|^2] \leq \left( \prod_{\ell=1}^k \sigma_m^\ell \right) \|\nabla P(\tilde{w}_0)\|^2.$$

As a corollary of our main theorem, it is easy to see that we can also obtain the convergence of *SARAH* Nguyen et al. (2017). Recall, from Remark B.2, that *SARAH* can be seen as a special case of *AI-SARAH* if, for all outer loops, $\alpha_{\min}^k = \alpha_{\max}^k = \alpha$. In this case, we can have

$$\sigma_m^k = \frac{1}{\mu\alpha(m+1)} + \frac{\alpha L_k^{\max}}{2 - \alpha L_k^{\max}}.$$

If we further assume that all functions $f_i$, $i \in [n]$, are $L$-smooth and do not take advantage of the local smoothness (in other words, do not use the working-set $\mathcal{W}_k$), then $L_k^{\max} = L$ for all $k \geq 1$. Then, with these restrictions, we have

$$\sigma_m = \sigma_m^k = \frac{1}{\mu\alpha(m+1)} + \frac{\alpha L}{2 - \alpha L} < 1.$$

As a result, Theorem B.9 guarantees the following linear convergence: $\mathbb{E}[\|\nabla P(\tilde{w}_k)\|^2] \leq (\sigma_m)^k \|\nabla P(\tilde{w}_0)\|^2$, which is exactly the convergence of classical *SARAH* provided in Nguyen et al. (2017).

## C  TECHNICAL PRELIMINARIES & PROOFS OF MAIN RESULTS

Let us start by presenting some important technical lemmas that will be later used for our main proofs.

### C.1  TECHNICAL PRELIMINARIES

**Lemma C.1.** *Nesterov (2003) Suppose that function $f$ is convex and $L$-Smooth in $C \subseteq \mathbb{R}^n$. Then for any $w, w' \in C$:*

$$\langle \nabla f(w) - \nabla f(w'), (w - w') \rangle \geq \frac{1}{L} \|\nabla f(w) - \nabla f(w')\|^2. \tag{12}$$

**Lemma C.2.** *Let Assumption 1 hold for all functions $f_i$ of problem (1). That is, let us assume that function $f_i$ is $L_k^i$-smooth $\forall i \in [n]$. Then, function $P(w) \overset{def}{=} \frac{1}{n}\sum_{i=1}^n f_i(w)$ is $\bar{L}_k$-smooth, where $\bar{L}_k = \frac{1}{n}\sum_{i=1}^n L_k^i$.*

*Proof.* For each function $f_i$, we have by definition of $L_k^i$-local smoothness,

$$f_i(x) \leq f_i(y) + \langle \nabla f_i(y), x - y \rangle + \frac{L_k^i}{2}\|x - y\|^2, \forall x, y \in \mathcal{W}_k.$$

Summing through all $i's$ and dividing by $n$, we get

$$P(x) \leq P(y) + \langle \nabla P(y), x - y \rangle + \frac{\bar{L}_k}{2}\|x - y\|^2, \forall x, y \in \mathcal{W}_k.$$

$\square$

The next Lemma was first proposed in Nguyen et al. (2017). We add it here with its proof for completeness and will use it later for our main theoretical result.

**Lemma C.3.** *Nguyen et al. (2017) Consider $v_t$ defined in (2). Then for any $t \geq 1$ in Algorithm 1, it holds that:*

$$\mathbb{E}[\|\nabla P(w_t) - v_t\|^2] = \sum_{j=1}^t \mathbb{E}[\|v_j - v_{j-1}\|^2] - \sum_{j=1}^t \mathbb{E}[\|\nabla P(w_j) - \nabla P(w_{j-1})\|^2]. \tag{13}$$

*Proof.* Let $\mathbb{E}_j$ denote the expectation by conditioning on the information $w_0, w_1, \ldots, w_j$ as well as $v_0, v_1, \ldots, v_{j-1}$. Then,

$$\begin{aligned}
\mathbb{E}_j[\|\nabla P(w_j) - v_j\|^2] &= \mathbb{E}_j\left[\|(\nabla P(w_{j-1}) - v_{j-1}) + (\nabla P(w_j) - \nabla P(w_{j-1})) - (v_j - v_{j-1})\|^2\right] \\
&= \mathbb{E}_j[\|\nabla P(w_{j-1}) - v_{j-1}\|^2] + \mathbb{E}_j[\|\nabla P(w_j) - \nabla P(w_{j-1})\|^2] + \mathbb{E}_j[\|v_j - v_{j-1}\|^2] \\
&\quad + 2\left(\nabla P(w_{j-1}) - v_{j-1}\right)^T \left(\nabla P(w_j) - \nabla P(w_{j-1})\right) \\
&\quad - 2\left(\nabla P(w_{j-1}) - v_{j-1}\right)^T \mathbb{E}_j[v_j - v_{j-1}] \\
&\quad - 2\left(\nabla P(w_j) - \nabla P(w_{j-1})\right)^T \mathbb{E}_j[v_j - v_{j-1}] \\
&= \mathbb{E}_j[\|\nabla P(w_{j-1}) - v_{j-1}\|^2] - \mathbb{E}_j[\|\nabla P(w_j) - \nabla P(w_{j-1})\|^2] + \mathbb{E}_j[\|v_j - v_{j-1}\|^2],
\end{aligned}$$

where the last equality follows from

$$\mathbb{E}_j[v_j - v_{j-1}] = \mathbb{E}_j[\nabla f_{i_j}(w_j) - \nabla f_{i_j}(w_{j-1})] = \nabla P(w_j) - \nabla P(w_{j-1}).$$

By taking expectation in the above expression, using the tower property, and summing over $j = 1, \ldots, t$, we obtain

$$\mathbb{E}[\|\nabla P(w_t) - v_t\|^2] = \sum_{j=1}^t \mathbb{E}[\|v_j - v_{j-1}\|^2] - \sum_{j=1}^t \mathbb{E}[\|\nabla P(w_j) - \nabla P(w_{j-1})\|^2].$$

$\square$

## C.2 Proofs of Lemmas and Theorems

For simplicity of notation, we use $|S| = 1$ in the following proofs, and a generalization to $|S| > 1$ is straightforward.

### C.2.1 Proof of Lemma B.7

By Assumption 1, Lemma C.2 and the update rule $w_t = w_{t-1} - \alpha_{t-1} v_{t-1}$ of Algorithm 1, we obtain:

$$
\begin{aligned}
P(w_t) &\leq P(w_{t-1}) - \alpha_{t-1}\langle \nabla P(w_{t-1}), v_{t-1}\rangle + \frac{\bar{L}_k}{2}\alpha_{t-1}^2 \|v_{t-1}\|^2 \\
&= P(w_{t-1}) - \frac{\alpha_{t-1}}{2}\|\nabla P(w_{t-1})\|^2 + \frac{\alpha_{t-1}}{2}\|\nabla P(w_{t-1}) - v_{t-1}\|^2 - \left(\frac{\alpha_{t-1}}{2} - \frac{\bar{L}_k}{2}\alpha_{t-1}^2\right)\|v_{t-1}\|^2,
\end{aligned}
$$

where, in the equality above, we use the fact that $\langle a, b\rangle = \frac{1}{2}(\|a\|^2 + \|b\|^2 - \|a - b\|^2)$.

By rearranging and using the lower and upper bounds of the step-size $\alpha_{t-1}$ in the outer loop $k$ ($\alpha_{\min}^k \leq \alpha_{t-1} \leq \alpha_{\max}^k$), we get:

$$
\frac{\alpha_{\min}^k}{2}\|\nabla P(w_{t-1})\|^2 \leq [P(w_{t-1}) - P(w_t)] + \frac{\alpha_{\max}^k}{2}\|\nabla P(w_{t-1}) - v_{t-1}\|^2 - \frac{\alpha_{t-1}}{2}\left(1 - \bar{L}_k\alpha_{t-1}\right)\|v_{t-1}\|^2.
$$

By assuming that $\alpha_{\max}^k \leq \frac{1}{\bar{L}_k}$, it holds that $\alpha_{t-1} \leq \frac{1}{\bar{L}_k}$ and $\left(1 - \bar{L}_k\alpha_{t-1}\right) \geq 0$, $\forall t \in [m]$. Thus,

$$
\frac{\alpha_{\min}^k}{2}\|\nabla P(w_{t-1})\|^2 \leq [P(w_{t-1}) - P(w_t)] + \frac{\alpha_{\max}^k}{2}\|\nabla P(w_{t-1}) - v_{t-1}\|^2 - \frac{\alpha_{\min}^k}{2}\left(1 - \bar{L}_k\alpha_{\max}^k\right)\|v_{t-1}\|^2.
$$

By taking expectations and multiplying both sides with $\frac{2}{\alpha_{\min}^k}$ :

$$
\begin{aligned}
\mathbb{E}[\|\nabla P(w_{t-1})\|^2] &\leq \frac{2}{\alpha_{\min}^k}[\mathbb{E}[P(w_{t-1})] - \mathbb{E}[P(w_t)]] + \frac{\alpha_{\max}^k}{\alpha_{\min}^k}\mathbb{E}[\|\nabla P(w_{t-1}) - v_{t-1}\|^2] - \left(1 - \bar{L}_k\alpha_{\max}^k\right)\mathbb{E}[\|v_{t-1}\|^2] \\
&\leq \frac{2}{\alpha_{\min}^k}[\mathbb{E}[P(w_{t-1})] - \mathbb{E}[P(w_t)]] + \frac{\alpha_{\max}^k}{\alpha_{\min}^k}\mathbb{E}[\|\nabla P(w_{t-1}) - v_{t-1}\|^2],
\end{aligned}
$$

where the last inequality holds as $\alpha_{\max}^k \leq \frac{1}{\bar{L}_k}$. Summing over $t = 1, 2, \ldots, m+1$, we have

$$
\begin{aligned}
\sum_{t=1}^{m+1}\mathbb{E}[\|\nabla P(w_{t-1})\|^2] &\leq \frac{2}{\alpha_{\min}^k}\sum_{t=1}^{m+1}\mathbb{E}[P(w_{t-1}) - P(w_t)] + \frac{\alpha_{\max}^k}{\alpha_{\min}^k}\sum_{t=1}^{m+1}\mathbb{E}[\|\nabla P(w_{t-1}) - v_{t-1}\|^2] \\
&= \frac{2}{\alpha_{\min}^k}\mathbb{E}[P(w_0) - P(w_{m+1})] + \frac{\alpha_{\max}^k}{\alpha_{\min}^k}\sum_{t=1}^{m+1}\mathbb{E}[\|\nabla P(w_{t-1}) - v_{t-1}\|^2] \\
&\leq \frac{2}{\alpha_{\min}^k}\mathbb{E}[P(w_0) - P(w_*)] + \frac{\alpha_{\max}^k}{\alpha_{\min}^k}\sum_{t=1}^{m+1}\mathbb{E}[\|\nabla P(w_{t-1}) - v_{t-1}\|^2],
\end{aligned}
$$

where the last inequality holds since $w^*$ is the global minimizer of $P$.

The last expression can be equivalently written as:

$$
\sum_{t=0}^{m}\mathbb{E}[\|\nabla P(w_t)\|^2] \leq \frac{2}{\alpha_{\min}^k}\mathbb{E}[P(w_0) - P(w_*)] + \frac{\alpha_{\max}^k}{\alpha_{\min}^k}\sum_{t=0}^{m}\mathbb{E}[\|\nabla P(w_t) - v_t\|^2],
$$

which completes the proof.

### C.2.2 Proof of Lemma B.8

$$
\begin{aligned}
\mathbb{E}_j\left[\|v_j\|^2\right] &\leq \mathbb{E}_j\left[\|v_{j-1} - \left(\nabla f_{i_j}(w_{j-1}) - \nabla f_{i_j}(w_j)\right)\|^2\right] \\
&= \|v_{j-1}\|^2 + \mathbb{E}_j\left[\|\nabla f_{i_j}(w_{j-1}) - \nabla f_{i_j}(w_j)\|^2\right] \\
&\quad - \mathbb{E}_j\left[\frac{2}{\alpha_{j-1}}\left\langle \nabla f_{i_t}(w_{j-1}) - \nabla f_{i_j}(w_j), w_{j-1} - w_j\right\rangle\right] \\
&\overset{(12)}{\leq} \|v_{j-1}\|^2 + \mathbb{E}_j\left[\|\nabla f_{i_j}(w_{j-1}) - \nabla f_{i_j}(w_j)\|^2\right] - \mathbb{E}_j\left[\frac{2}{\alpha_{j-1}L_k^{i_j}}\|\nabla f_{i_j}(w_{j-1}) - \nabla f_{i_j}(w_j)\|^2\right].
\end{aligned}
$$

For each outer loop $k$, it holds that $\alpha_{j-1} \leq \alpha_{\max}^k$ and $L_k^i \leq L_k^{\max}$. Thus,

$$
\begin{aligned}
\mathbb{E}_j[\|v_j\|^2] &\leq \|v_{j-1}\|^2 + \mathbb{E}_j\left[\|\nabla f_{i_j}(w_{j-1}) - \nabla f_{i_j}(w_j)\|^2\right] - \frac{2}{\alpha_{\max}^k L_k^{\max}}\mathbb{E}_j\left[\|\nabla f_{i_j}(w_{j-1}) - \nabla f_{i_j}(w_j)\|^2\right] \\
&= \|v_{j-1}\|^2 + \left(1 - \frac{2}{\alpha_{\max}^k L_k^{\max}}\right)\mathbb{E}_j\left[\|\nabla f_{i_j}(w_{j-1}) - \nabla f_{i_j}(w_j)\|^2\right] \\
&= \|v_{j-1}\|^2 + \left(1 - \frac{2}{\alpha_{\max}^k L_k^{\max}}\right)\mathbb{E}_j\left[\|v_j - v_{j-1}\|^2\right].
\end{aligned}
$$

By rearranging, taking expectations again, and assuming that $\alpha_{\max}^k < 2/L_k^{\max}$:

$$
\mathbb{E}[\|v_j - v_{j-1}\|^2] \leq \left(\frac{\alpha_{\max}^k L_k^{\max}}{2 - \alpha_{\max}^k L_k^{\max}}\right)\left[\mathbb{E}[\|v_{j-1}\|^2] - \mathbb{E}[\|v_j\|^2]\right].
$$

By summing the above inequality over $j = 1, \ldots, t$ ($t \geq 1$), we have:

$$
\begin{aligned}
\sum_{j=1}^{t}\mathbb{E}[\|v_j - v_{j-1}\|^2] &\leq \left(\frac{\alpha_{\max}^k L_k^{\max}}{2 - \alpha_{\max}^k L_k^{\max}}\right)\sum_{j=1}^{t}\left[\|v_{j-1}\|^2 - \|v_j\|^2\right] \\
&\leq \left(\frac{\alpha_{\max}^k L_k^{\max}}{2 - \alpha_{\max}^k L_k^{\max}}\right)\left[\mathbb{E}[\|v_0\|^2] - \mathbb{E}[\|v_t\|^2]\right]. \quad (14)
\end{aligned}
$$

Now, by using Lemma C.3, we obtain:

$$
\begin{aligned}
\mathbb{E}[\|\nabla P(w_t) - v_t\|^2] &\overset{(13)}{\leq} \sum_{j=1}^{t}\mathbb{E}\left[\|v_j - v_{j-1}\|^2\right] \\
&\overset{(14)}{\leq} \left(\frac{\alpha_{\max}^k L_k^{\max}}{2 - \alpha_{\max}^k L_k^{\max}}\right)\left[\mathbb{E}[\|v_0\|^2] - \mathbb{E}[\|v_t\|^2]\right] \\
&\leq \left(\frac{\alpha_{\max}^k L_k^{\max}}{2 - \alpha_{\max}^k L_k^{\max}}\right)\mathbb{E}[\|v_0\|^2]. \quad (15)
\end{aligned}
$$

### C.2.3 PROOF OF THEOREM B.9

*Proof.* Since $v_0 = \nabla P(w_0)$ implies $\|\nabla P(w_0) - v_0\|^2 = 0$, then by Lemma B.8, we obtain:

$$
\sum_{t=0}^{m}\mathbb{E}[\|\nabla P(w_t) - v_t\|^2] \leq \left(\frac{m\alpha_{\max}^k L_k^{\max}}{2 - \alpha_{\max}^k L_k^{\max}}\right)\mathbb{E}[\|v_0\|^2]. \quad (16)
$$

Combine this with Lemma B.7, we have that:

$$
\begin{aligned}
\sum_{t=0}^{m}\mathbb{E}[\|\nabla P(w_t)\|^2] &\leq \frac{2}{\alpha_{\min}^k}\mathbb{E}[P(w_0) - P(w_*)] + \frac{\alpha_{\max}^k}{\alpha_{\min}^k}\sum_{t=0}^{m}\mathbb{E}[\|\nabla P(w_t) - v_t\|^2] \\
&\overset{(16)}{\leq} \frac{2}{\alpha_{\min}^k}\mathbb{E}[P(w_0) - P(w_*)] + \frac{\alpha_{\max}^k}{\alpha_{\min}^k}\left(\frac{m\alpha_{\max}^k L_k^{\max}}{2 - \alpha_{\max}^k L_k^{\max}}\right)\mathbb{E}[\|v_0\|^2] \quad (17)
\end{aligned}
$$

Since we are considering one outer iteration, with $k \geq 1$, we have $v_0 = \nabla P(w_0) = \nabla P(\tilde{w}_{k-1})$ and $\tilde{w}_k = w_t$, where $t$ is drawn uniformly at random from $\{0, 1, \ldots, m\}$. Therefore, the following holds,

$$
\begin{aligned}
\mathbb{E}[\|\nabla P(\tilde{w}_k)\|^2] &= \frac{1}{m+1}\sum_{t=0}^{m}\mathbb{E}[\|\nabla P(w_t)\|^2] \\
&\overset{(17)}{\leq} \frac{2}{\alpha_{\min}^k(m+1)}\mathbb{E}[P(\tilde{w}_{k-1}) - P(w_*)] + \frac{\alpha_{\max}^k}{\alpha_{\min}^k}\left(\frac{\alpha_{\max}^k L_k^{\max}}{2 - \alpha_{\max}^k L_k^{\max}}\right)\mathbb{E}[\|\nabla P(\tilde{w}_{k-1})\|^2] \\
&\leq \left(\frac{1}{\mu\alpha_{\min}^k(m+1)} + \frac{\alpha_{\max}^k}{\alpha_{\min}^k}\left(\frac{\alpha_{\max}^k L_k^{\max}}{2 - \alpha_{\max}^k L_k^{\max}}\right)\right)\mathbb{E}[\|\nabla P(\tilde{w}_{k-1})\|^2].
\end{aligned}
$$

Let us use $\sigma_m^k = \frac{1}{\mu \alpha_{\min}^k (m+1)} + \frac{\alpha_{\max}^k}{\alpha_{\min}^k} \cdot \frac{\alpha_{\max}^k L_k^{\max}}{2 - \alpha_{\max}^k L_k^{\max}}$, then the above expression can be written as:

$$\mathbb{E}[\|\nabla P(\tilde{w}_k)\|^2] \leq \sigma_m^k \mathbb{E}[\|\nabla P(\tilde{w}_{k-1})\|^2].$$

By expanding the recurrence, we obtain:

$$\mathbb{E}[\|\nabla P(\tilde{w}_k)\|^2] \leq \left( \prod_{\ell=1}^k \sigma_m^\ell \right) \|\nabla P(\tilde{w}_0)\|^2.$$

This completes the proof. $\qquad\square$

