# OpenReview forum: "AI-SARAH: Adaptive and Implicit Stochastic Recursive Gradient Methods"
_ICLR.cc/2022/Conference — ICLR 2022 Submitted_

### Official Review · Reviewer_G45K · 2021-10-16

**Correctness:** 3
**Technical Novelty And Significance:** 2
**Empirical Novelty And Significance:** 2
**Recommendation:** 6
**Confidence:** 4

**Main Review:**

Pros:

(1) The combination of adaptive tuning-free stepsize + SARAH is novel. Its motivation is very clear.

(2) The experiments show that AI-SARAH outperforms some main-stream stochastic optimizers.

Concerns:

(1) My major concern is that both the theorem and the experiments only involve convex optimization, while nonconvex optimization is more popular in machine learning, and SARAH achieves near-optimal sample complexity for nonconvex optimization. For convex optimization, SVRG and SAGA [2] have better sample complexity than SARAH. Hence, I suggest adding both theorem and experiments for nonconvex optimization.

(2) Also, for convex optimization, usually we obtain convergence rate of function value gap $f(w_t)-f(w^*)$ or point distance $\|w_t-w^*\|$, while the authors only obtain convergence rate of gradient norm which is weaker and usually obtained for nonconvex optimization.

(3) I suggest adding theoretical results of sample complexity (The number of computing function value and derivatives) and compare with that of SARAH for nonconvex optimization, since most works on SGD and variance reduction techniques (including SARAH) have sample complexity results. For the convex optimization, the complexity result can be directly obtained from Theorem 1.

**I reject this paper mainly due to the above three concerns. Once they are solved, I will change to accept.**

(4) The algorithm is claimed as "First-order", while $\xi'(0)$ and $\xi''(0)$ require the second and the third order derivatives of $P$ respectively. Hence, the sample complexity result also needs to involve the number of evaluations of the second and the third order derivatives.

(5) There is an unclear point in Algorithm 1. If $k>0$ and $t=1$, then the line 14 is implemented using the undefined $\delta_0^k$. What is the value of $\delta_0^k$?

Minor comments:

(1) It is better to add y-axis labels to Figures 1c, 1d and 8. For example, The y-axis label of Figure 8 may be "stepsize".

(2) In line 10 of Algorithm 1, you could add "using $\xi_t$ defined in (4)", and in (4), I think you could still use $f_{S_t}$ since that looks the same simple as $f_{i_t}$ and guides us to correctly use Algorithm 1. Then, when explaining (4) by the example of quadratic function, you could let $S_t=\{i_t\}$ for simplicity.

(3) For nonconvex optimization, you might use STORM algorithm [1] instead of SARAH, since both of them achieve the near-optimal sample complexity for nonconvex finite-sum optimization, but STORM does not require full gradient.

(4) At the beginning of page 5, use "Let" (capitalized) in "let us focus on a simple quadratic function".

(5) You might compare with SAGA [2], another important variance reduction technique as well in your experiment.

(6) The authors may add intuition of the line 17 of Algorithm, i.e., the reason for using $\alpha_{\max}$.

(7) In the final paragraph of Section 1.2, all the variance reduced work I read use $w_{t+1}=w_t-\alpha_t v_t$, with either constant stepsize $\alpha_t=\alpha$ or predetermined diminishing stepsize. Also "allow use" could be "allow to use".

References:

[1] Cutkosky, Ashok, and Francesco Orabona. Momentum-based variance reduction in non-convex sgd. ArXiv:1905.10018 (2019).

[2] Defazio, Aaron, Francis Bach, and Simon Lacoste-Julien. SAGA: A fast incremental gradient method with support for non-strongly convex composite objectives. Advances in neural information processing systems. 2014.

**Summary Of The Paper:**

This paper proposes a novel algorithm by using adaptive tuning-free stepsize in SARAH, obtains its convergence rate in convex optimization, and demonstrates its empirical superiority by logistic regression experiments.

**Summary Of The Review:**

I reject this paper mainly due to the above concerns 1-3. Once they are solved, I will change to accept.

---

> ### Author Response · Authors · 2021-11-18
> **Response to the comments of Reviewer G45K ("Concerns")**
>
> $\textbf{Please see the following for response to "concerns"}$
>
> $\textbf{A1)}$ Solving nonconvex problems is out of the scope of this paper. Indeed, the original SARAH paper [1] is a good example, whose focus is on the strongly convex settings. That said, we acknowledge that solving nonconvex problems, especially training deep neural networks, is of a great level of interest in the field. And, we are continuously working on extending AI-SARAH's territories in this setting. In this paper, we make a solid step in developing a truly adaptive and tune-free practical algorithm and it delivers a very strong performance in solving convex problems.
>
> $\textbf{A2)}$ Please note that the convergence with respect to the squared norm (or the full gradient, i.e., $||\nabla P(w_k)||^2$) is used in the original SARAH paper [1], and we adopt the same scheme in this paper. Also note that, by showing the convergence of $||\nabla P(w_k)||^2$, we show that the algorithm converges to a stationary point. For (strongly) convex functions, the stationary point is the global minimum, and thus it is a valid approach in analysis.
>
> $\textbf{A3)}$ We appreciate the recommendation of adding the complexity analysis in the paper. In this paper, our main contribution is to develop the practical algorithm, AI-SARAH (Algorithm 1). It is truly adaptive and tune-free with very strong empirical performance. For the theoretical analysis, we basically focus on answering the design questions, that is $\textbf{(1)}$ does the algorithm even converge by using local Lipschitz smoothness instead of a constant step-size? $\textbf{(2)}$ if it does, what is the benefit of using it for step-size?  The analysis answers these two questions well and thus serves the purpose.
>
> $\textbf{A4)}$ We appreciate the careful review of the technical details. It requires the second and third derivatives of $f_{S_t}$ as being prescribed in $v_t$. As we addressed clearly in the paper (Section 3.2 and Appendix A), it requires two additional backward passes. For fair comparison with other algorithms, we show wall clock time in Figures of section 4 and others in Appendix. For theoretical analysis, we do not explicitly address how to solve the sub-problem, and that is because of the reason mentioned earlier.
>
> $\textbf{A5)}$ The algorithm does not start the iteration at $t=0$. Instead, it starts with $t=1$ for the inner loop (as shown on Line 7 of Algorithm 1), and $\delta^0_1$ is shown on Line 12 of Algorithm 1. We do not set initial upper-bound before the first iteration (at $k=0,t=1$), i.e., $\alpha_{max} = \infty$ on Line 3 of Algorithm 1.
>
> [1] Lam M. Nguyen, Jie Liu, Katya Scheinberg, and Martin Takác. Sarah: A novel method for machine ˇ learning problems using stochastic recursive gradient.

---

> > ### Comment · Reviewer_G45K · 2021-11-23
> > **Reviewer G45K's 1st reply**
> >
> > Hello, authors,
> > Thank you for your elaborate response.
> > I am persuaded by most of your answers. Just two comments:
> >
> > (1) A5: Since you have already implemented SAGA, you could add to the experiments.
> >
> > (2) Before I increase my rating to 6, could you answer the following question.
> > A5: When $t=k=1$, Algorithm 1 computes $\delta_1^1=\beta\delta_0^1+(1-\beta)/\tilde{\alpha}_{t-1}$ which requires $\delta_0^1$. However, the line 12 defines $\delta_1^0$ instead of $\delta_0^1$. Could you explain that?

---

> > > ### Author Response · Authors · 2021-11-23
> > > **Response to 1st reply**
> > >
> > > Thank you very much for your response! We have to say it was indeed a great catch, and we apologize for neglecting the typo/error before the deadline for the pdf revision.
> > >
> > > To correct the error, we will add, before Line 23 of Algorithm 1, $$\textbf{Set} \\; \delta^{k+1}_0 = \delta^k_t. \Leftarrow \text{as we do carry this quantity throughout the entire running process and not resetting the upper-bound per outer loop.}$$ To also correct the typo in Algorithm 1, on Line 11, we will change it to $$\textbf{if} \\; k=1 \textbf{ and } t=1 \textbf{ then }. \Leftarrow \text{as the algorithm starts from k=1}.$$
> > >
> > > To further clarify the updating schema, the algorithm starts with an initial $\delta$, which is shown on Line 12; after the very first iteration, the algorithm will only use Line 14 for all subsequent iterations.
> > >
> > > Again, we really appreciate your attention to the details, which will make the presentation precise and error-free. We hope it answer the question. Please let us know if there is any further comments on this item. Thank you!

---

> > > > ### Comment · Reviewer_G45K · 2021-11-23
> > > > **Reviewer G45K's 2nd reply**
> > > >
> > > > Thank you for your correction.
> > > > My concerns are addressed and I have increased my rating to 6.

---

> > > > > ### Author Response · Authors · 2021-11-23
> > > > > **Response to 2nd reply**
> > > > >
> > > > > Thank you for increasing the score. We really appreciate your time and efforts in a careful review of our paper!

---

> ### Author Response · Authors · 2021-11-18
> **Response to the comments of Reviewer G45K ("Minor Comments")**
>
> $\textbf{Thank you for the comments listed under the "minor comments".}$ We really appreciate the efforts in pointing out such detailed advice. We revised the paper based on your comments for most of the items. For others, we list our feedback here:
>
> $\textbf{A3)}$ For the extension to nonconvex optimization, we certainly will take action on the advice.
>
> $\textbf{A5)}$ We have SAGA in the introduction and the reference. Prior to many iterations of finishing the writing of this paper, we had SAGA implemented for the experiments. We thought it sufficed to compare AI-SARAH with the algorithms included in the paper, but we can certainly add SAGA to the experiment if it is considered as the bottleneck to this paper.
>
> $\textbf{A6})$ We had the intuition of using $\alpha_{max}$ in Section 3.1: "Clearly, if a step-size in the algorithm is selected as $1/\bar L_t$, then a harmonic mean of the sequence of the step-size, computed for various component functions could serve as a good upper-bound on the step-size computed in the algorithm." Per your advice, we added more intuitions into Appendix A.

---

> ### Author Response · Authors · 2021-11-18
> **Thank you for your comments on strenght**
>
> $\textbf{We thank reviewers for the compliments on the strengh:}$ $\textbf{(1)}$ tune-free algorithm; $\textbf{(2)}$ adaptive step-size; $\textbf{(3)}$ clear motivation; $\textbf{(4)}$ strong performance in practice. It is indeed the contribution that we hope to make through this paper.
>
> We hope we were able to address the concerns that the reviewer had, and we kindly ask the reviewer to re-assess the merit of our paper.

---

### Official Review · Reviewer_4uKg · 2021-11-01

**Correctness:** 4
**Technical Novelty And Significance:** 2
**Empirical Novelty And Significance:** 2
**Recommendation:** 5
**Confidence:** 5

**Main Review:**

The authors presented a parameter-free and hence tunning-free variant of the SARAH variance reduction method. An extensive empirical study of the algorithm has been done and the result shows some advantages of the proposed method compared to other state-of-the-art methods that may need parameter tuning. This is the strength of the result.

However, the weakness of the paper is that, though the authors do show some theoretical results, there is a big gap between the theory and the implemented algorithm.



**Summary Of The Paper:**

In this paper, the authors proposed a SARAH-type variance reduced gradient method that adaptively and automatically selects the stepsize.
In each step, the authors propose to do an approximate line-search of the currently sampled function $f_{i_t}$ (or $f_{S_t}$) to estimate the local Lipschitz constant of the sampled function. To estimate the Lipschitz constant of the true objective function (the summation), the authors propose to do an exponential moving average of the estimated local Lipschitz constants.

Extensive experiments are done in solving convex problems, and the comparison with other state-of-the-art first-order methods is also carefully presented. The authors also presented some theoretical guarantees of the algorithm with certain modifications.


**Summary Of The Review:**

The reviewer does appreciate the efforts in the empirical study. However, the reviewer has a few comments about the gap between theory and practice.

1. The strength of the proposed algorithm is "tuning-free" and "parameter-free". However, to theoretically guarantee the convergence, the author actually requires knowing the local Lipschitz constant in the set $\mathcal{W}_k$, see Eq(6).

2. The authors argue that as the iterations approach the optimal solution, the curvature may tend to be flatter. Therefore, the global Lipschitz constant may be much larger than the local Lipschitz constant, resulting in a more conservative stepsize, i.e., the stepsize $1/L_{global}$ may be much smaller than the stepsize $1/L_{local}$. However, since $\alpha_{max}^k$ is already $1/2L_{local}$, doing a line search only results in smaller stepsizes. This contradicts the philosophy of using larger stepsize when possible. Therefore, the theory does not explain the strength of the algorithm. Moreover, if $1/L_{local}$ is known, why not directly apply SARAH with this local parameter?

3. There is a small flaw in the logic of the proof. First, the authors define the set \mathcal{W}_k:= {w:||w- \tilde w_{k-1}|| \leq m\alpha_{max}^k||v_0||}. The authors argued that $\mathcal{W}_k$ is the set where all the iterations of the $k$-th epoch should lie within. This only happens when all the later gradient estimators $||v_t||\leq ||v_0||$ almost surely, otherwise the proof wouldn't work. However, the authors are not able to show this because all the descent results are in the sense of expectation instead of almost surely.

4. With $\alpha_{max}^k = 1/2L_{local}$, the analysis is more or less the same as the original SARAH algorithm, with little modification. Therefore, the current analysis is not very interesting to the reviewer. However, a much more interesting thing is how well the proposed procedure approximates the local gradient, this has never been addressed in the existing works. However, with the current assumptions on $\alpha_{max}^k$, this challenge is left undone.

-----------------------------------------------------------------------------------------------------------------------------------------------------------------------------------
After the rebutal, the authors have cleared my question in comment #3. The issue stated in this comment can be cleared with a few more explanation.

---

> ### Author Response · Authors · 2021-11-18
> **Response to the comments of Reviewer 4uKg (1-2,4)**
>
> $\textbf{Please see the following for response to the weakness listed underneath "the summary".  }$
>
> $\textbf{A1)}$ AI-SARAH, as a practical algorithm, does not require any knowledge of local geometry or hyper-parameters, and thus we say it is tune-free and fully adaptive algorithm. In theoretical analysis, we are trying to answer two questions: $\textbf{(1)}$ can we even use local Lipschitz smoothness in the first place; $\textbf{(2)}$ if we can, is it possible to achieve faster convergence than SARAH. To answer these two questions, we show that, given the knowledge of local Lipschitz smoothness, the algorithm does converge. And, it could be faster than SARAH if local geometry permits. The analysis validates the idea of using local geometry, and AI-SARAH in practice can automatically estimate the local Lipschitz smoothness without prior knowledge.
>
> $\textbf{A2)}$ We appreciate the careful review of the details. However, we feel there is a need to clarify a few things.
>
> $\textbf{A2.1)}$ First of all, AI-SARAH is a practical algorithm and the motivation (in section 2) of using local Lipschitz smoothness is the phenomenon that the reviewer mentioned.
>
> $\textbf{A2.2)}$ In practice, we verify the phenomenon and results can be found in Figure 1 (c) of section 2, Figure 8 of section 4, and Figures 22-23 of Appendix A.
>
> $\textbf{A2.3)}$ Our practical algorithm AI-SARAH does not use line-search or need knowledge of local Lipschitz smoothness to determine a step-size. Instead, it estimates the local Lipschitz smoothness by solving the sub-problem (see section 3.1) and determines the appropriate step-size automatically.
>
> $\textbf{A2.4)}$ In practice, it is almost always the case that local Lipschitz smoothness is unknown; AI-SARAH estimates it by investing two additional backward passes (see section 3.2). Thus, it is not appropriate for one to assume knowing $L_{local}$ in practice.
>
> $\textbf{A2.5)}$ The theoretical analysis focuses on answering the question that is yet to be answered for SARAH type of algorithm: can local Lipschitz smoothness be used to determine a step-size for the algorithm to converge.
>
> $\textbf{A2.6)}$ In theoretical analysis, we use $\alpha_{max}$ (and $\alpha_{min}$ to restrict the step-size and facilitate the proof. It is a common practice when dealing with technical proofs on adaptive step-size for gradient-based algorithms.
>
> $\textbf{A4)}$ Thank you for your advice, and could you please be more specific on the "approximates the local gradient"? AI-SARAH does compute $\nabla f_{S_t}$ if it was referred to as "local gradient" in the comments. For the comments of the analysis, please see the above response for details. To summarize, the analysis answers the design question of "can a stochastic recursive algorithm use local Lipschitz smoothness to determine step-size", and that serves the purpose. The main contribution is on the practical algorithm AI-SARAH (Algorithm 1), where we show how to estimate local smoothness, the strong performance in practice, and it is truly adaptive and tune-free.

---

> > ### Author Response · Authors · 2021-11-23
> > **Response to the comments of Reviewer 4uKg (comment #3)**
> >
> > $\textbf{A3).}$ Thank you for your comments on the technical details, but $\textbf{there is no flaw in the logic of the proof}$. We did not include it in the paper as we thought it was trivial. We can certainly add it to the paper if the reviewer believes it is beneficial to the audience.
> >
> > Here, we argue that it is always the case that
> > $\\|v_t\\| \leq \\|v_0\\|$ and all the iterates of the $k$-th outer loop lie within the set
> > $\mathcal W_k := \\{w \in \mathcal R^d \\; | \\; \\|\tilde w_{k-1} - w\\| \leq m \cdot \alpha_{max}^k \\|v_0\\|\\}.$
> >
> > $\textbf{(1)}$. First of all, we can show that, as long as $w_{t-1}, w_t \in \mathcal W_k$, $\\|v_t\\| \leq \\|v_{t-1}\\|$ deterministically.  By definition of $v_t$ and the assumption of the main theorem that $\alpha_{t-1} \leq \alpha_{max}^k \leq \frac{2}{L_{max}^k}$. Given $w_t, w_{t-1} \in \mathcal W_k$,
> > \begin{align*}
> > ||v_t||^2 &= || v_{t-1} - (\nabla f_{i_t} (w_{t-1}) - \nabla f_{i_t} (w_t))||^2 \\\\
> > &= || v_{t-1} ||^2 + || \nabla f_{i_t} (w_{t-1}) - \nabla f_{i_t} (w_t) ||^2 - \frac{2}{\alpha_{t-1}} \langle \nabla f_{i_t}(w_{t-1}) - \nabla f_{i_t}(w_t), w_{t-1} - w_t \rangle\\\\
> > &\leq || v_{t-1} ||^2 + || \nabla f_{i_t} (w_{t-1}) - \nabla f_{i_t} (w_t) ||^2 - L_{max}^k \langle \nabla f_{i_t}(w_{t-1}) - \nabla f_{i_t}(w_t), w_{t-1} - w_t \rangle\\\\
> > & \leq || v_{t-1} ||^2.
> > \end{align*}
> > The last inequality holds as $f_{i_t}$ is convex and smooth with parameter $L_{max}^k$ on $\mathcal W_k$.
> >
> > $\textbf{(2)}$. Now, we can show the main results and note that $\tilde w_{k-1} = w_0$.
> >
> > By induction, at $j=1$, $\\|w_0 - w_1\\| = \alpha_0 \\|v_0\\| \leq m \cdot \alpha_{max}^k \\|v_0\\|$, and thus $w_1 \in \mathcal W_k$. By $\textbf{(1)}$, $\\|v_1\\| \leq \\|v_0\\|$.
> >
> > Assume for $1<j\leq t$, $w_j \in \mathcal W_k$, then by $\textbf{(1)}$, $\\|v_j\\| \leq \\|v_{j-1}\\|$ and thus $\\|v_j\\| \leq \\|v_0\\|$.
> >
> > Then, at $j=t+1$, we have
> > $\\|w_0 - w_{t+1}\\| = \\|\sum_{i=1}^{t+1} \alpha_{i-1} v_{i-1}\\| \leq \alpha_{max}^k \sum_{i=1}^{t+1}\\|v_{i-1}\\| \leq m \cdot \alpha_{max}^k \\|v_0\\| \Rightarrow w_{t+1} \in \mathcal W_k \text{ and } \\|v_{t+1}\\| \leq \\|v_0\\|.$
> >
> > Therefore, by $\textbf{(1)}$ and $\textbf{(2)}$, we show the desired results.

---

> > > ### Comment · Reviewer_4uKg · 2021-11-25
> > > **Response to comment #3.**
> > >
> > > Thank you for the clarification. The response does make sense.

---

> ### Author Response · Authors · 2021-11-18
> **Thank you for your comments on the strength**
>
> $\textbf{We thank the reviewer for the comments on the strength:}$ $\textbf{(1)}$ tune-free algorithm; $\textbf{(2)}$ extensive experiments and strong performance. Indeed, our contribution is on proposing the tune-free and fully adaptive algorithm in practice and demonstrating it has very strong performance through empirical study.
>
> We hope we were able to address the concerns that the reviewer had, and we kindly ask the reviewer to re-assess the merit of our paper.

---

> ### Author Response · Authors · 2021-11-26
> **follow-up with reviewer 4uKg**
>
> Thank you for your feedback! It seems we are not able to upload a new pdf here, but we have made updates in the paper to include $\textbf{A3}$.
>
> As the updated comment says “the issue stated in this comment can be cleared with a few more explanations” and we haven't received responses for 1-2,4, we would like to ask if there are any concerns/clarifications that we can address/provide. We would be more than happy to discuss any of those!
>
> Thank you!

---

> ### Author Response · Authors · 2021-11-29
> **Response to Reviewer 4uKg**
>
> Dear Reviewer 4uKg,
>
> Thank you very much for your review!!! We believe that all your questions were answered in our rebuttal and response. As we mentioned in the previous response, we will certainly add $\textbf{A3}$ to the paper, as well as details in line with our response to value your input on "more explanations". We believe these would be minor editings and can be handled easily in a camera-ready paper if accepted.
>
> We kindly ask you to re-evaluate our paper and re-consider the score.
>
> Thank you!
>
> Authors

---

### Official Review · Reviewer_Y92t · 2021-11-02

**Correctness:** 3
**Technical Novelty And Significance:** 2
**Empirical Novelty And Significance:** Not applicable
**Recommendation:** 5
**Confidence:** 4

**Details Of Ethics Concerns:**

NA, optimization algorithm.

**Main Review:**

Strengths:

1: This paper proposes a practical variant of SARAH with faster convergence by exploring the local geometry of the stochastic functions.

2: Extensive experiments are conducted to verify the effectiveness and efficiency of the proposed algorithm.

3: This paper is well-organized and well written with clear language and structures. The background knowledge is presented and the related papers are cited.


Weaknesses:

1: The contribution of SARAH over SVRG is mainly on convex functions rather than strongly convex functions. However, this paper only analyzes the case of the strongly convex function for AI-SARAH. As a variant of SARAH, I believe theoretical results on the general convex function are required.

2: In convex optimization, the convergence results are usually evaluated with regard to the function value rather than the squared norm of the gradient. However, the proposed AI-SARAH is usually slower than the compared methods with regard to the function value.

3: To tune the hyperparameter of SARAH and SVRG, in my personal experience, we do not really need 5000 runs to find a good hyperparameter. The authors may need to explain more on that.


**Summary Of The Paper:**

This paper studies the stochastic recursive gradient method for finite-sum problems. The proposed approach AI-SARAH is a practical variant of SARAH by exploring the local geometry of the stochastic functions. Specifically, in each iteration of the inner loop, AI-SARAH estimates the parameter of local Lipschitz smoothness by solving a sub-problem so that can use a larger step size for faster convergence when approaching the optimum.

**Summary Of The Review:**

Although the authors make some contributions in this paper, the weaknesses above are very critical to evaluating the contributions of the proposed method. I would like to vote for 'reject' if they are not addressed well.

After the discussion, some of the concerns are solved. I will increase the score to 5.

---

> ### Author Response · Authors · 2021-11-18
> **Response to comments of Reviewer Y92t**
>
> $\textbf{Please see the following for response to the weakness.  }$
>
> $\textbf{A1)}$ The original SARAH paper [1] mainly shows a linear convergence rate under a strongly convex assumption in the theoretical analysis and demonstrates the empirical merit with regularized logistic regression, which is strongly convex. Thus, for AI-SARAH, we focus on deriving the results in the similar manner as the main results of SARAH, and extend the empirical study to cover both regularized and non-regularized cases, where the latter can be non-strongly convex. (As a side note, the novelty of SARAH given the presence of SVRG is mainly on designing a biased estimator of stochastic gradient and its linear convergence, i.e., $\|v_t\|^2 \rightarrow 0$, within an inner loop. This is the main contribution of SARAH.)
>
> $\textbf{A2)}$ Please note that the convergence with respect to the squared norm (or the full gradient, i.e., $||\nabla P(w_k)||^2$) is used in the original SARAH paper [1], and we adopt the same scheme in this paper. Also note that, by showing the convergence of $||\nabla P(w_k)||^2$, we show that the algorithm converges to a stationary point. For (strongly) convex functions, the stationary point is the global minimum, and thus it is a valid approach in analysis. In empirical study, it is possible that AI-SARAH is slower than other methods in some cases to minimize the functions. But, it is also possible that AI-SARAH is faster than other methods. For example, for $\textit{real-sim}$ and $\textit{ijcnn1}$ in Figure 6, AI-SARAH could be faster than SARAH.
>
> $\textbf{A3)}$ As mentioned in Section 4 and Appendix A, we compare AI-SARAH with ADAM, SGD w/ momentum, SVRG, SARAH and SARAH+. There are $\approx 5000$ runs for all configurations of hyper-parameters for all five algorithms that AI-SARAH is comparing with, not just SARAH and SVRG. We also explain the tuning plan with extensive details in text (see Section 4 and Appendix B) and table (see Table 3). The idea is to compare AI-SARAH that is tune-free with the fine-tuned versions of the other algorithms. For this purpose, it is necessary to perform extensive hyper-parameter search. For example, for SARAH and SVRG, we choose 16 loop sizes and 10 step-sizes, each of the choices are reasonable and could be potentially the best among all choices. As a results, there are 160 configurations yielded for SARAH (or SVRG). As a common practice, we run each configuration for 5 random seeds to take into account the impact of randomness (in both initialization and mini-batch data stream).
>
> [1] Lam M. Nguyen, Jie Liu, Katya Scheinberg, and Martin Takác. Sarah: A novel method for machine ˇ
> learning problems using stochastic recursive gradient.

---

> > ### Comment · Reviewer_Y92t · 2021-11-23
> > **Reply to the Response**
> >
> > Hi authors, thank you for your response! Some of my questions are addressed. I still have some concerns with regard to the replies:
> >
> > 1: I still cannot find any convergence results of the proposed AI-SARAH on the general convex function, which is a class of important functions in machine learning. However, the theoretical analysis on the general convex function has been conducted for all the compared methods SGD, Adam, SVRG, SARAH.
> >
> > 2: In the original paper of SARAH, which is four years later than the paper of SVRG and is an improved version of SVRG, SARAH or SARAH+ is one of the fastest methods for all the compared methods, such as SGD, Adam, SVRG, and at least perform similarly to SVRG. However, in this paper, SARAH and SARAH+ are the slowest methods for all the methods, such as SGD, Adam, SVRG. I believe it may need some explanations for that. Even for the case mentioned for ijcnn1 and real-sim, the SARAH method is much slower than other methods. Meanwhile, AI-SARAH is slower than most of the compared methods.

---

> > > ### Author Response · Authors · 2021-11-23
> > > **Response to the two remaining issues**
> > >
> > > Dear reviewer, the deadline for our response is coming soon, so let us give a quick reply.
> > >
> > > 1. Yes, methods mentioned by you (SGD, Adam, SVRG, SARAH) have convergence in a general convex setting. But let us mention that for example original SVRG paper https://proceedings.neurips.cc/paper/2013/file/ac1dd209cbcc5e5d1c6e28598e8cbbe8-Paper.pdf
> > > did not analyze a general convex setting, JUST a strongly convex one.
> > > We believe that extending our result to convex should be possible, we just haven't had enough time to do it during the rebuttal period (two weeks was not sufficient for us to do so due to many other commitments). Also, if the level set of the function would be bounded (diameter R), one can add a small regularization ~ \frac{\epsilon}{4 R^2} \|x - x_0\|^2 and optimize that (strongly convex function) to get approximate solution for the original problem
> > > (see  Theorem 3 of Nesterov paper
> > > https://cdn.uclouvain.be/public/Exports%20reddot/core/documents/coredp2010_2web.pdf )
> > > Our main goal was really to just demonstrate the new concept here and have the practical algorithm.
> > >
> > > 2. The issue related to experiments in SARAH paper: it is hard to judge the experiments in SARAH paper. It seems to be 4 years old paper when many researchers haven't run experiments multiple times. They are not showing the average and standard deviations. All algorithms mentioned above are stochastic. For example, in AI-SARAH in Figure 4 one can see the variance of results for various algorithms. We are not sure what you mean that AI-SARAH is the slowest? It is slower per iteration Fig.3, but overall, because of the adaptive learning rate, it will be much faster (see e.g., Fig. 4, were we put x-axis as time and show that AI-Sarah is getter better solution for given time budget without any tuning).
> > >
> > > Does it explain the concern?

---

> > > > ### Author Response · Authors · 2021-11-26
> > > > **follow-up on reviewer's remaining questions**
> > > >
> > > > Thank you for discussing your remaining questions with us. We hope the above response from us answered your questions. Please let us know if there are any questions to the above response.
> > > >
> > > > Thank you!

---

> ### Author Response · Authors · 2021-11-18
> **Thank you for comments on strength**
>
> $\textbf{We thank the reviewer for the compliments on the strength:}$
> $\textbf{(1)}$ Exploration of local geometry drives a faster convergence than SARAH; $\textbf{(2)}$ extensive experiments and strong performance of AI-SARAH; $\textbf{(3)}$ well organized & written paper and clear structure & background knowledge.
>
> Indeed, as we mentioned in the paper, we believe our contribution is the development of a fully adaptive and tune-free practical algorithm that delivers a strong empirical performance. We kindly ask the reviewer to re-evaluate the merits of the paper.

---

> ### Author Response · Authors · 2021-11-26
> **Response from authors**
>
> Thank you for your consideration! We hope the latest response from us answers the questions you had previously.
>
> As the updated comments says “some of the concerns are solved”, we would like to ask if there are any concerns/clarifications that we can follow-up with you. We would be more than happy to discuss any of those!
>
> Thank you!

---

> ### Author Response · Authors · 2021-11-29
> **Response to Reviewer Y92t**
>
> Dear Reviewer Y92t,
>
> Thank you very much for your review and consideration!!! We believe, at this moment, all your questions, as well as your "remaining issues" were answered in our response. Although we believe adding a general convex case is only a minor improvement as addressed in our response, we will certainly add it to value your input. And, it can be handled easily for a camera-ready paper if accepted.
>
> We kindly ask you to re-assess the paper based on the response and re-consider the score.
>
> Thank you!
>
> Authors

---

### Official Review · Reviewer_CpvK · 2021-11-02

**Correctness:** 3
**Technical Novelty And Significance:** 2
**Empirical Novelty And Significance:** 3
**Recommendation:** 5
**Confidence:** 4

**Main Review:**

- I cannot see it explained how to pick $\alpha_{max}^k$, In Lemmas B.7 and B.8 they depend on some Lipschitz constant depending on the set in eq. (11) how to estimate these?
- The authors talk about using Newton method to solve the sub-problem, but how does this affects the analysis explicitly?
- What about adaptivity to strong convexity? While it is easy to know the strong convexity for the regularizers, it can be quite difficult to estimate the strong convexity of the loss function ("data-term"). The authors say the algorithm is "fully adaptive". Probably,it should be explained that this "full adaptivity" is not to the strong convexity.


For the experiments, I understand the authors point: even though each iteration of AI-SARAH can be expensive because of the local Lipschitz constant estimation, it doesn't require tuning and the other methods require tuning. The authors state that they do "fine-tuning" for all the other methods, which takes a lot of time and therefore the overall cost of AI-SARAH is reasonable. My question is: what happens when the authors do a "rough tuning"? Is the improvement of the other methods is significant between the "rough tuning" and "fine tuning"? Maybe "fine tuning" isn't necessary for these methods to be competitive and the overall time for these algorithms can be reduced this way. It would be nice to see a tradeoff of these algorithms in terms of the accuracy/total time/budget of tuning.


- Why is sparsity of the datasets mentioned? Does the algorithm use this information in any way?

Overall, I am concerned that the writing is not too transparent to show this fact about the difference of analyzed and implemented algorithms. I find the discussions before Section 3.4 too long and repetitive whereas Section 3.4 is too quick and unclear for the readers why all these modifications are being done for analysis.


While I do appreciate the value of extensive practical comparisons, for such a well-studied class of problems (smooth and strongly convex), I expect a smaller gap between the theory and practice within the paper. Therefore, I think the authors should either implement the algorithm they analyze or analyze an algorithm closer to the one they implement. Moreover, if the paper wants to make a claim more on the practical side, then I am curious what happens with "a rough and systematic tuning" for other algorithms.


**Summary Of The Paper:**

This paper proposes a new variance reduced algorithm for solving smooth and strongly convex optimization. The authors build on SARAH and propose AI-SARAH which aims to estimate the local Lipschitz smoothness parameters on the fly. The authors provide theoretical analysis for a modified version of AI-SARAH and provide extensive practical experiments for AI-SARAH.


**Summary Of The Review:**

I really appreciate the promise of the paper and the direction pursued by the authors. However, I have some concerns regarding the theory-practice mismatch within the paper. Unfortunately, until Section 3.4, the writing is as if the authors will analyze AI-SARAH which is implemented in practice, however Section 3.4 introduces many modifications on the algorithm. On this front, it is not explained if the analyzed algorithm does really work well in practice or how much theory can be shown for the implemented algorithm.

I am willing to increase my score if the authors reply to my questions in a satisfactory way.

---

> ### Author Response · Authors · 2021-11-18
> **Response to comments of Reviewer CpvK**
>
> First and foremost, we would like to respond to the comments listed as $\textbf{the last three paragraphs of "Main Review" and the first paragraph of "the summary"}$.
>
> $\textbf{(1)}$ The main contribution of the paper is on the development of our practical algorithm, AI-SARAH (Algorithm 1). It can automatically estimate the local Lipschitz smoothness and compute the step-size. It is fully adaptive, tune-free, and able to deliver a very strong performance in practice. Sections 2 and 3 are very important as it explains the motivation and approach of AI-SARAH in detail; more details are deferred in Appendix A.
>
> $\textbf{(2)}$ The theoretical analysis basically is trying to answer the questions: Can we even use local Lipschitz smoothness to design the algorithm? If so, what is the benefit? Then, through the analysis, we are able to show the convergence by using local Lipschitz smoothness, and show that a faster convergence is possible than original SARAH [1]. The analysis verifies the design idea of AI-SARAH (Algorithm 1), which is to use local Lipschitz smoothness to derive adaptive step-size.
>
> $\textbf{(3)}$ While we would like to get clarification on "a rough and systematic tuning for other algorithms", what we conducted in experiments is a systematic hyper-parameter tuning. The purpose is to present the best-tuned version of these algorithms and be able to compare AI-SARAH with them. We admit that it is impossible to search for infinite space for hyper-parameters, but the tuning details/plan (as shown in Section 4 and Table 3 in Appendix A) is reasonable and sufficient for the other algorithms. And, with sufficient searching of hyper-parameters, we can achieve the conclusion in the empirical study: AI-SARAH has very strong performance when compared with other algorithms.
>
> [1] Lam M. Nguyen, Jie Liu, Katya Scheinberg, and Martin Takác. Sarah: A novel method for machine ˇ learning problems using stochastic recursive gradient.

---

> > ### Comment · Reviewer_CpvK · 2021-11-28
> > **Post-rebuttal comments**
> >
> > I see that my main concern (a big gap between theory and practice) is also pointed out by other reviewers. It seems the authors' reply is to pose a vague question for theory: "Can we even use local Lipschitz smoothness to design the algorithm?". Unfortunately, such "is it possible"-type questions are much easier to answer when many additional assumptions are made for analysis on top of the implemented algorithm, which seems to be the path that the authors picked. I recommend the authors to analyze the algorithm they implement with the good practical performance.
> >
> > By "a rough and systematic tuning", I mean using less number of possibilities for the loops and the seeds. The authors mention they use 5 random seeds, 16 loop size, 10 step size. What happens with 8 loop size 5 step size (with the same upper and lower bounds), 2 random seed etc. I think it is necessary to include such "tuning with different budgets" for other methods to be really fair.
> >
> > Overall, I agree that the promise of the paper is interesting, but the current results aren't sufficient for this promise. For their revisions, I recommend the authors to close the gap between their analyzed and implemented algorithm as much as possible and also to include experiments with different tuning budgets for the other methods for a fair and more convincing comparison.

---

> > > ### Author Response · Authors · 2021-11-29
> > > **Response to Reviewer CpvK**
> > >
> > > $\textbf{Thank you for your feedback on our rebuttal!!! }$
> > > First of all, we would like to point out that in the rebuttal, while we are still waiting for other reviewers' follow-up, we were able to address most of the concerns on theoretical analysis from other reviewers, and it resulted in raising the scores by two reviewers by far.
> > >
> > > $\textbf{(1)}$
> > >
> > > For your concern on the aforementioned gap, we note that our goal is to propose a practical algorithm as clearly written in the paper and addressed in the rebuttal.
> > >
> > > $\textbf{(a)}$ First, we argue that the question is not vague. It is indeed a precise and significant question that we are trying to investigate. It is precise as it is directly related to the derivation of step-size. It is significant as if we couldn't find the answer or if a local Lipschitz smoothness is not usable for determining a dynamic step-size, there would not exist our practical algorithm, AI-SARAH. And, such a question was not answered prior to our work. We acknowledge that, compared with developing a practical algorithm, tune-free and fully adaptive, it is a bit easier to reach our conclusion in theory that we can indeed use local Lipschitz smoothness.
> > >
> > > $\textbf{(b)}$ Following our argument in $\textbf{(a)}$, in theory, we do not discuss how to get the local Lipschitz smoothness, how to solve the sub-problem, and how to update an upper-bound, etc. The reason is that these are not the goal of the theory as we addressed above and in the rebuttal. Besides, they cannot be answered perfectly in theory, and it is either not necessary or not possible to do so. For example, how to solve the sub-problem exactly (given the sub-problem is not convex)? how to get the local Lipschitz smoothness precisely without evaluating the Hessian (on a ray or within a ball)? However, not knowing these answers in theory does not block our way from designing our practical algorithm as the major question has been answered: we can use local Lipschitz smoothness to design dynamic step-size.
> > >
> > > $\textbf{(c)}$ We acknowledge that it is always limited to how much one can show in the theory with absolute rigor for every piece of the elements in developing the practical algorithm. However, the items mentioned in $\textbf{(b)}$ are exactly the ones we try to solve as part of the algorithm development. And, for AI-SARAH to be tune-free, adaptive and competitive, each subsection underneath Section 3 plays a different but critical role.
> > >
> > > $\textbf{(2)}$
> > >
> > > Thank you for your clarification. We kindly disagree with you as
> > >
> > > $\textbf{(a)}$ First, we note that our conclusion is that $\textbf{AI-SARAH (Algorithm 1)}$ has a strong and competitive performance $\textbf{even when compared with other algorithms with fine-tuned hyper-parameters}$. We could certainly make fewer efforts as you mentioned to reach the conclusion. However, the evidence to support the conclusion would be weaker if compared with what we have in the paper. As we addressed in the rebuttal, we would like to make sure we compare AI-SARAH with the best possible hyper-parameters for other algorithms, so we can be certain of our conclusion. In Figures 5-7, we presented the main performance metrics, where we only presented one single best-tuned hyper-parameters for each of the others algorithms that AI-SARAH is comparing with.
> > >
> > > $\textbf{(b)}$ We are not intended to make any conclusion that how much efforts one should spend on tuning the other algorithms. In practice, one could spend more or fewer efforts than this paper (or even zero effort if just randomly picking hyper-parameters). But, it does not change the fact that these algorithms require tuning for strong performance. On the other hand, this paper shows that AI-SARAH is tune-free and produces good performance. On top of that, this paper shows that even if one fined-tuned the other algorithms, AI-SARAH would be very competitive. Then, the competitive performance of AI-SARAH would still hold when compared with the other algorithms which are "roughly and systematically tuned".
> > >
> > > We hope our response addresses your concerns. Given the timeline of the discussion period, we would really appreciate any follow-up discussion with you. In the meantime, we kindly request a re-evaluation of our paper based on the rebuttal and response.
> > >
> > > Thank you!

---

> ### Author Response · Authors · 2021-11-18
> **Response to the comments of Reviewer CpvK (Weakness shown as bullets)**
>
> $\textbf{A1)}$ In the analysis of Appendix B, $\alpha_{max}$ is defined in Lemmas B.7 and B.8, and it depends on the work set $W_k$ as the set prescribes $\bar{L_k}$. As mentioned above, it is not the goal to estimate Lipschitz smoothness in the theoretical analysis. In practice, AI-SARAH (Algorithm 1) can efficiently estimate it, and we refer it to section 3.
>
> $\textbf{A2)}$ Thank you for your careful review of the details. The Newton method we adopt is to approximately solve the sub-problem in AI-SARAH (Algorithm 1). The analysis does not address the part of "how to estimate the local Lipschitz smoothness". Instead, it is to verify that, given the knowledge of local smoothness, the algorithm can converge. Thus, it validifies the idea of deriving adaptive step-size using local geometry in AI-SARAH (Algorithm 1).
>
> $\textbf{A3)}$ Please note that the adaptivity is on the adaptive step-size. To be precise, the step-size is automatically determined by having AI-SARAH (Algorithm 1) adapt to the local Lipschitz smoothness. In the design of the algorithm, there is no involvement of strong convexity parameters (see Sections 2 & 3) in step-size determination.
>
> $\textbf{A4)}$ We display the sparsity as part of the datasets summary, and it is a common practice to give such information. To answer the second question, the design of the algorithm does not depend on the sparsity of the datasets. However, in practice, given a hugely sparsed dataset (like some of the datasets we used in the paper), implementing algorithms by leveraging sparsity is a common practice. To be specific, we implemented all algorithms in python based on our sparse layer implementation for PyTorch. Thus, for the experiments, all algorithms are fed by sparse matrices as input instead of dense matrices.

---

> ### Author Response · Authors · 2021-11-18
> **Thank you for your review**
>
> $\textbf{We thank Reviewer CpvK for the detailed and constructive review comments}$, we hope that our response addresses the concern that the reviewer had in the comments, and kindly ask the reviewer to reassess the merits of the paper.

---

> ### Author Response · Authors · 2021-11-26
> **follow-up with Reviewer CpvK**
>
> Thank you for your initial review of our paper!
>
> We hope our response addressed/answered the concerns/questions that you had in the initial review. If there are any questions to our response, please let us know. We would be more than happy to discuss any of those with you!
>
> Thank you!

---

> ### Author Response · Authors · 2021-11-30
> **Follow-up response to Reviewer CpvK**
>
> Dear Reviewer CpvK,
>
> Thank you very much for your review and feedback! We believe we were able to address most of your concerns, if not all, in our response! We sincerely thank you for giving us the opportunity to explain the main idea of the paper! And, we wish we could have more time for discussion with you! We certainly value your feedback and will add detailed explanations in line with our response to a camera-ready paper if accepted!
>
> We kindly ask you to re-evaluate our paper and re-consider the score based on our response!
>
> Thank you!
>
> Authors

---

### Author Response · Authors · 2021-11-23
**General Response**

We thank all the reviewers for the constructive feedback and comments. We were pleased to see reviewers found the strength of the paper as being able to propose a tune-free and adaptive algorithm in practice (Algorithm 1) in a well-organized way, and all reviewers complimented/appreciated our empirical study, which indeed demonstrates the competitive performance of the algorithm.

As a number of reviewers commented on the technical details/theoretical analysis, we hope we were able to address those concerns in our response. We kindly ask the reviewers to re-evaluate our paper based on our response. And, we would be more than happy to have further discussions on any of your concerns.

---

> ### Author Response · Authors · 2021-12-01
> **Thank you, Reviewers.**
>
> Dear Reviewers,
>
> Thank you so much for your time and efforts in reviewing our paper! We were happy to get your feedback and be able to discuss your concerns. We thank all of you for your constructive feedback, and we appreciate the raise of the scores by two Reviewers.
>
> We believe AI-SARAH, with very strong empirical performance backed by extensive experiments, could serve as a stepping stone for future research in the exciting direction of fully-adaptive algorithms.
>
> At this moment, we believe we have already addressed all of your concerns. But, given we can still post messages, we would be more than happy to answer any of the further questions that you might have.
>
> At the same time, If you had no further questions, we would really appreciate it if you can reconsider your score.
>
> Thank you!
> Authors

---

### Decision · Program_Chairs · 2022-01-20

**Decision:**

Reject

**Comment:**

This paper presents a variant of SARAH, which employs the stochastic recursive gradient and adjustable step-size based on local geometry. The main concerns about this paper include (1) the empirical comparison with other algorithms might not be fair (which is arguable); and (2)  the theorem proved in the paper is for a simplified algorithm rather than the algorithm used in the experiments. Even after author response and reviewer discussion, this paper does not gather sufficient support from the reviewers. Thus I recommend rejection.